# Large Language Models to Diffusion Finetuning

Edoardo Cetin [1]  Tianyu Zhao [1]  Yujin Tang [1]

## Abstract

We propose a new finetuning method to provide pre-trained large language models (LMs) the ability to scale test-time compute through the diffusion framework. By increasing the number of diffusion steps, we show our finetuned models achieve monotonically increasing accuracy, directly translating to improved performance across downstream tasks. Furthermore, our finetuned models can expertly answer questions on specific topics by integrating powerful guidance techniques, and autonomously determine the compute required for a given problem by leveraging adaptive ODE solvers. Our method is applicable to any foundation model pre-trained with cross-entropy and does not modify any of its original weights, fully preserving its strong single-step generation capabilities. We show our method can be more effective and is fully compatible with traditional finetuning and search approaches, introducing an orthogonal new direction to unify the strengths of the autoregressive and diffusion frameworks.

## 1. Introduction

The scalability of autoregressive large language models (LMs) is a pivotal component of the current generation of foundation models (Team et al., 2023; Achiam et al., 2023; Dubey et al., 2024). However, despite their unprecedented capabilities, LMs inherently lack many valuable properties that could be expected of an "artificial general intelligence," such as the ability to scale computation for their most critical decisions (Sutton, 2019). Efforts to address this limitation primarily focused on eliciting more nuanced responses through prompting and targeted searches over the space of possible completions (Feng et al., 2023; Kumar et al., 2024; Trinh et al., 2024; Jaech et al., 2024), anchoring the reasoning process in the space of generated tokens.

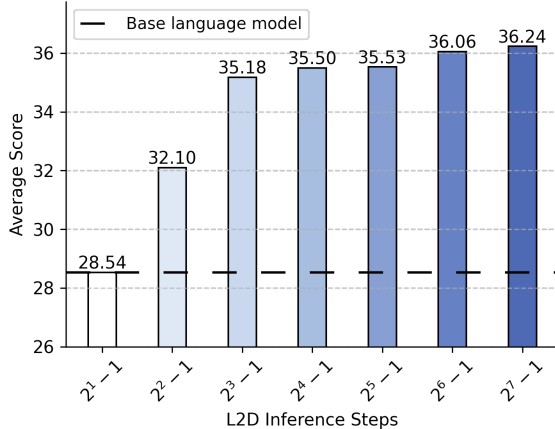

Figure 1. **Test-time compute scaling with L2D.** Our framework empowers LMs with the scaling properties of diffusion, yielding increasingly higher inference performance with additional steps.

Established as the predominant approach in visual domains, the diffusion framework offers properties that appear particularly complementary to the LM paradigm (Sohl-Dickstein et al., 2015; Song & Ermon, 2019; Ho et al., 2020; Dhariwal & Nichol, 2021; Peluchetti, 2023; Esser et al., 2024). For instance, the iterative nature of diffusion allows to adaptively scale compute to the difficulty of a specific task or any level of accuracy demanded by the user, regardless of the generated output's length. However, despite these useful properties, diffusion models trained for language currently lag significantly behind their autoregressive counterparts (Lou et al., 2024; Gat et al., 2024; Gulrajani & Hashimoto, 2024) putting into question their inductive bias and scalability when applied to this highly relevant domain.

In this work, we aim to unite the strengths of these frameworks by introducing LM to Diffusion (L2D): a new finetuning method powering pre-trained LMs with the scaling properties and potential of diffusion (Karras et al., 2022; Uehara et al., 2025). Rather than learning a diffusion model from scratch, our method harnesses the large amount of "system 1" understanding efficiently acquired during autoregressive pre-training by casting LMs as single-step diffusions. Then, by introducing a small fraction of new parameters – comparable to modern parameter-efficient approaches (Hu et al., 2021) – we imbue the model with a new set of multi-step "reasoning" skills, the ability to scale computation on-demand, and the potential to incorporate powerful guidance techniques (Ho & Salimans, 2022), all without compromising its original

[1]Sakana AI, Tokyo, Japan. Correspondence to: Edoardo Cetin <edo@sakana.ai>, Tianyu Zhao <tianyu@sakana.ai>, Yujin Tang <yujintang@sakana.ai>.

*Proceedings of the 42nd International Conference on Machine Learning*, Vancouver, Canada. PMLR 267, 2025. Copyright 2025 by the author(s).

single-step capabilities.

In summary, our technical contributions are the following:

- We introduce L2D, a new finetuning method to power LMs with the scaling properties of diffusion, combining key strengths from these two frameworks.

- We show that L2D significantly improves four different LMs on math, coding, and a variety of reasoning tasks; and that its benefits can be both superior and complementary to traditional finetuning and search.

- We demonstrate that L2D allows to scale performance with additional compute, while opening the door to LMs equipped with autonomous per-token scaling and powerful diffusion guidance techniques.

We provide our full code[1] to facilitate future advances in developing new scalable foundation models with diffusion.

## 2. Gaussian Diffusion for LM Finetuning

In this section, we describe the key components of our L2D framework. In particular, we provide details about the considered diffusion formulation, together with our designed training and inference approaches. Although each of the following subsections offers a concise introduction to the concepts and modern practices of diffusion and language modeling, we refer to recent work (Nakkiran et al., 2024; Lipman et al., 2024) and Section 5 for more comprehensive resources. We conclude the section explaining how our design decisions make L2D a natural extension to modern language modeling aimed to complement rather than supersede the autoregressive framework.

### 2.1. Gaussian Diffusion

Gaussian diffusion decomposes the problem of generating new samples from a target unknown distribution $p^*$ from a source distribution $q := N(0, I)$ over multiple "simpler" steps. The Gaussian diffusion decomposition effectively reuses the intermediate information computed in the model's attempts in each previous step. These subsequent diffusion steps can be seen as a discretization of a continuous "denoising" process from $t = 0$ to time $t = 1$, over which the model is tasked to transform samples from $q$ to $p^*$. All intermediate distributions along the denoising process are defined by a corresponding corruption process, mixing target data points $x_1 \sim p^*$ with noise from $q$ to produce $x_t \sim p_t$:

$$x_t = \alpha_t x_1 + \beta_t x_0, \quad \text{where} \quad x_0 \sim N(0, I). \quad (1)$$

Here, the schedules $\alpha_t$ and $\beta_t$ are defined as monotonic functions with $\alpha_0 = \beta_1 = 0$ and $\alpha_1 = \beta_0 = 1$, satisfying the constraints such that $p_0 := q$ and $p_1 := p^*$.

[1]https://github.com/SakanaAI/L2D

Neural networks (NNs) in single-step generative modeling solely rely on an external source of pure randomness to generate new samples from scratch. In contrast, the goal of diffusion is to learn a neural network $f_\theta$ conditioned on samples from each $p_t$ and tasked with solving the simpler problem of generating new samples from lower nearby noise levels $p_{t+\Delta_t}$. Thus, effectively splitting the challenge of learning and generating new samples in multiple steps, which can be scaled based on computational availability.

### 2.2. L2D Parametrization and Training Formulation

An effective choice of loss to train diffusion models is simply to predict the values in the uncorrupted target datapoints from $p_1$ (i.e., $p^*$) given the partial information contained at each corruption level $\hat{x} = f_\theta(x_t, t)$. When $p_1$ is a distribution over a continuous domain, this is commonly done by using a simple mean squared regression loss on all timesteps $t$, as popularized by the DDPM algorithm (Ho et al., 2020):

$$L^{L2}(\theta) = \mathbb{E}_{t,x_0,x_1} \left[ ||x_1 - f_\theta(x_t, t)||_2^2 \right]. \quad (2)$$

Another key design decision for diffusion is the choice of schedules $\alpha_t$ and $\beta_t$, which define the denoising process that $f_\theta$ will be learning. This is one of the most significant choices for continuous diffusion models, affecting all aspects of both training and inference dynamics (Nichol & Dhariwal, 2021; Esser et al., 2024). In our work, we employ the schedules $\alpha_t = t$ and $\beta_t = (1-t)\sigma$, where $\sigma$ is a hyperparameter linearly scaling the signal-to-noise ratio for all timesteps between $p_1$ and $p_0$ within the samples $x_t \sim p_t$. This choice is closely tied to the rectified flow matching schedules (Liu et al., 2022), which have been shown to possess particularly desirable "straightening" properties for diffusion (Lee et al., 2024; Lipman et al., 2024) and have been widely adopted in the recent diffusion literature (Esser et al., 2024). To ease our notation and make this connection explicit, we absorb the hyper-parameter $\sigma$ in the standard deviation of our base distribution $p_0 := N(0, \sigma^2 I)$, which simplifies our schedules to $\alpha_t = t$ and $\beta_t = (1-t)$.

Unlike for the continuous case, language modeling operates over a target distribution $p_1$ defined on a finite vocabulary table $V$, where to each index $y \in 1, \ldots, |V|$ there corresponds a token embedding $x \in \mathbb{R}^d$. This key difference is one of the main reasons that diffusion in language modeling is yet to have a predominant recipe with several recent approaches even exploring alternative diffusion formulations over the discrete space of vocabulary indices $y$ (Austin et al., 2021a; Lou et al., 2024; Gat et al., 2024). In this work, we choose to still diffuse over the token embeddings $x$, as in standard continuous diffusion, but do not employ an MSE loss as done by Li et al. (2022). Instead, we learn our diffusion model with a simple cross-entropy loss, establishing a direct connection to traditional single-step language modeling. In particular, given a token $x_1$ indexed by label $y$ sampled

**Algorithm 1** *Diffusion language modeling predictions*
___
1: **Input** diffusion model $f_\theta$, context $c$, budget $T$
2: Initialize $t \leftarrow 0$, $\Delta_t \leftarrow 1/(T-1)$
3: Sample $x_t \sim N(0, \sigma^2 I)$
4: **for** $i = 1, 2, ..., T-1$ **do**
5:     Sample $y_t \sim f_\theta(x_t, t, c)$
6:     Set $\hat{x} \leftarrow V_{y_t}$
7:     Compute $dx_t = \frac{\hat{x} - x_t}{1-t}$
8:     Update $t \leftarrow t + \Delta_t$, $x_t \leftarrow x_t + \Delta_t \times dx_t$
9: **end for**
10: **Return** $y \sim f_\theta(x_1, 1, c)$
___

along with a context of preceding tokens $c$ from the target data distribution $p_1$, our diffusion loss is formulated as:

$$L^{CE}(\theta) = -\mathbb{E}_{x_0, x_1, t}\left[\log\left(f_\theta(x_t, t, c)_y\right)\right], \quad \text{where}$$
$$x_0 \sim N(0, \sigma^2 I), \quad x_1 = V_y \sim p_1, \quad (3)$$
$$t \sim U[0, 1] \quad \text{and} \quad x_t = tx_1 + (1-t)x_0.$$

This formulation allows our diffusion network $f_\theta$ to still predict $|V|$ logits over the vocabulary tokens, just like a standard language model, while leveraging partial information about the next sequence token provided by $x_t$. Despite its simplicity, this choice still enables our diffusion process to draw a continuous trajectory during inference, similar to traditional diffusion models with continuous outputs as explained by Dieleman et al. (2022) and detailed below.

### 2.3. L2D Inference Formulation

To generate new samples with a traditional continuous diffusion model, an effective approach is to use the predictions $\hat{x}$ from $f_\theta(x_t, t)$ to construct an ODE that preserves the marginal distribution $p_t$ at each timestep $t$ (Song et al., 2020a;b). While many such valid ODEs exist for a single diffusion process, we adopt the formulation from Liu et al. (2022), which is designed to yield a constant expected velocity along the denoising trajectory at each timestep $t$:

$$dx_t = \frac{\hat{x} - x_t}{1-t}. \quad (4)$$

The denoising process can then start at $t = 0$ by drawing $x_t$ from pure noise and be performed over a sequence of steps where previous predictions are reused to bring $x_t$ to a lower noise level at $t + \Delta_t$ toward the direction $dx_t$. In the simplest case, this process amounts to Euler integration where $x_{t+\Delta_t} = x_t + \Delta_t \times dx_t$. However, any ODE solver can be employed with constant or adaptive costs given by fixed discretization levels $\Delta_t$ or adaptive accuracy requirements.

Given our parameterization of $f_\theta$, outputting categorical probabilities over the vocabulary, its predictions cannot be directly used to obtain $dx_t$ as with continuous diffusion. However, as shown by Dieleman et al. (2022), we can use

these probabilities together with the vocabulary embeddings stored in $V$ to estimate $\hat{x}$ for any valid velocity (in our case, defined in Equation 4). While Dieleman et al. (2022) takes $\hat{x}$ as the weighted average over the embeddings, we instead use the probabilities predicted by $f_\theta(x_t, t, c)$ to sample an individual $\hat{x} \in V$ at each diffusion step $t$. Although the expectation of these two estimates matches, we note our choice reintroduces some stochasticity into the denoising trajectory traced by the ODE. In practice, we find this stochasticity beneficial to better harness some of the self-correcting properties of the diffusion framework, which Karras et al. (2022) showed might be limited in fully deterministic inference formulations. We summarize this next-token prediction procedure with our sampling approach (lines 5-6), Euler integration, and a budget of $T$ total steps in Algorithm 1.

### 2.4. LMs as Single-step Diffusion Models

Our choices in designing L2D establish a clear connection with the traditional LM framework. As detailed above, training a diffusion model with Equation 3 can be interpreted as standard next-token prediction where the model is provided with an additional "diffusion token" $x_t$ containing some amount of knowledge about the target $y$, ranging from no information ($t = 0$) to perfect information ($t = 1$). Therefore, LMs are essentially trained with an equivalent prediction objective to L2D's when $t = 0$, where $x_t$ is entirely uncorrelated with the target $y$. Similarly, inference following Algorithm 1 involves iteratively sampling increasingly accurate next tokens $\hat{x}$ from the model's logits up to a sampling budget $T$. Thus, traditional LM inference can be again viewed as a special case of this procedure with $T = 1$, where only the model's first sample is used to predict $y$.

The purpose of these design choices is that L2D aims to extend pre-trained LMs via a finetuning approach, rather than learning new models from scratch. While fully adopting diffusion training from the start might appear more general, we argue this risks losing some of the training scalability and powerful inductive biases inherent to traditional autoregressive modeling which led to their wide establishment in the language domain (Allen-Zhu & Li, 2023a;b). Furthermore, L2D directly enables leveraging the extensive "system 1" understanding (Kahneman, 2013) already encoded in open foundation models. In fact, by building on their existing capabilities we avoid the prohibitive costs required in past attempts to match their performance with diffusion.

## 3. L2D Implementation

We design our L2D implementation as a modular extension for pre-trained transformers to efficiently harness the multi-step scaling capabilities of diffusion while preserving their original single-step generative power. To achieve this, L2D introduces a parallel "diffusion" path to their archi-

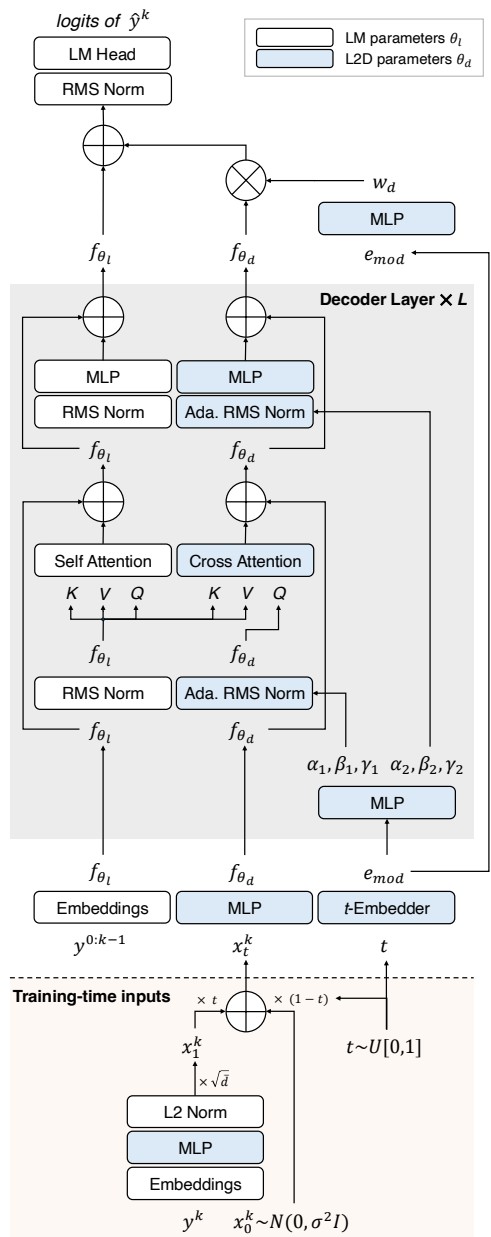

*Figure 2.* **L2D LMs overview.** Training-time sampling of diffusion tokens $x_t^{1:k}$ (bottom) and architecture diagram for L2D (top).

tecture, where the hidden representation of the diffusion token $x_t$ is propagated, affecting the frozen "main" LM path only at the final layer. In this section, we provide details about each specific L2D component, highlighting how our choices ensure scalability and efficiency advantages over prior designs. To accompany our explanations, we show an overview of the L2D pipeline illustrating transformer architectures augmented with our framework in Figure 2.

### 3.1. Diffusion Path Parametrization

**Structure and initialization.** We process the diffusion tokens $x_t$ within a separate parallel path to the LM's original

architecture. This choice allows us to optimize only a subset of the model's parameters with no risk of losing its original ability to process the "uncorrupted" tokens in the context $c$. We implement the diffusion path, denoted $f_{\theta_d}$, with a transformer architecture and the same number of blocks as the main path $f_{\theta_l}$, each comprising a subset of its layers (from the MLP blocks and the query layers in self-attention). Moreover, to make the most of the pre-trained LM's knowledge, all layers in the diffusion path are also initialized with the weights from $\theta_l$, similarly to Zhang et al. (2023). In practice, we find this initialization enables fast and inexpensive training, allowing us to optimize the diffusion path with simple low-rank adaptation (LoRA, Hu et al. 2021). Furthermore, this approach greatly minimizes L2D's memory overhead, as it requires us only to store the small LoRA modules by reusing the LM's original weights in both $\theta_d$ and $\theta_l$.

**Diffusion path components.** The transformer blocks in the diffusion path comprise a sequence of residual MLP and cross-attention modules. While the MLP modules follow the same structure as the corresponding modules in $f_{\theta_l}$, the cross-attention modules exclusively parameterize query and output linear layers. In particular, during cross-attention, the diffusion token $x_t^k$ for target token $y^k$ attends over all previous keys and values already computed from the corresponding self-attention module in $f_{\theta_l}$. We only integrate the information processed in $f_\theta$ back to the main path after all blocks, right before the LM's linear head. Specifically, we merge the two paths with an element-wise weighted sum $f_{\theta_l} + w_d f_{\theta_d}$ where the rescaled latents of diffusion token $x_t^k$ are added to the latents of the previous token $x^{k-1}$.

**Properties and advantages.** Our design choices have several key advantages over prior diffusion architectures targeted for multi-token generation (Li et al., 2022; Dieleman et al., 2022). During inference, by saving the latent representation from $f_{\theta_l}$ together with the KV cache, we only need to compute the output of the main path once for each generated token, no matter the number of diffusion steps. Furthermore, as the diffusion token for the $k$-th target only affects the main path at the previous position, we can fully parallelize training across the sequence batch dimension, sampling timesteps $t_1 \ldots t_K$ and diffusion tokens $x_{t_1}^1 \ldots x_{t_K}^K$ independently. By doing this, we greatly mitigate the variance of the diffusion optimization objective, efficiently obtaining independent diffusion losses for all $K$ sequence positions for each sampled input context $x^0 \ldots x^{K-1}$ in the data batches.

### 3.2. L2D Conditioning

**Diffusion space vocabulary.** To condition $f_{\theta_d}$, we construct the vocabulary containing the discrete set of token embeddings for the diffusion path $x \in V$ from the pre-trained token vocabulary of the base LM, denoted $V^l$. In particu-

lar, we learn a linear mapping $W_v \in \mathbb{R}^{\bar{d} \times d}$ to convert each pre-trained embedding $V_y^l$ to an efficient lower-dimensional embedding in $\mathbb{R}^{\bar{d}}$, later rescaled to a fixed norm $\sqrt{\bar{d}}$:

$$V_y = \sqrt{\bar{d}} \frac{W_v V_y^l}{||W_v V_y^l||_2}, \quad \text{for all} \ \ y = 1, \dots |V|. \quad (5)$$

This normalization step is required to avoid the magnitude of the tokens in $V$ growing unboundedly to minimize the corruption effects from the sampled noises $x_0 \sim N(0, \sigma^2 I)$ while training with Equation 3. Instead, as proposed by Dieleman et al. (2022), this approach will make the token embeddings in $V$ naturally spread out, which will lead to their distribution possessing unit variance in each component across the data manifold. Lastly, we use a small 2-layer "translation module" at the beginning of the diffusion path, mapping back the diffusion tokens embeddings to $\mathbb{R}^d$ for compatibility with the transformer blocks in $f_{\theta_d}$.

**Timestep conditioning.** We condition the diffusion path on the current timestep $t \in [0, 1]$ in three distinct ways. First, based on established practices from the modern diffusion literature, we extract sinusoidal features from $t$ and process them with a small network to output shift and scale parameters for all layer normalizations in $f_{\theta_d}$. Second, following Peebles & Xie (2023), we parametrize additional time-conditioned element-wise rescalings which we apply before summing back the residuals from each transformer block. Third, we make final use of the timestep embeddings to condition the last element-wise weighting term $w_d$ used to scale the outputs of the diffusion path $f_{\theta_d}$. However, rather than making this weight the output of a network $w_{\theta_d}(t)$, like in the first two cases, we shift $w_d$ with the value of $w_{\theta_d}(0)$:

$$w_d(t) = w_{\theta_d}(t) - w_{\theta_d}(0). \quad (6)$$

The main direct consequence of this parametrization is that the diffusion path will always be multiplied with zeros at $t = 0$, leaving the original output of $f_{\theta_l}$ unchanged. Thus, this practice ensures that L2D will never trade off the powerful single-step capabilities of the pre-trained LM when $x_t$ is pure noise, and provides a strong inductive bias for the diffusion path to increasingly affect predictions as $t$ grows to 1 and $x_t$ contains more past compute and knowledge.

**Classifier-free guidance.** Finally, we can effectively condition L2D models on additional contextual information about a task or a dataset through classifier-free guidance (Ho & Salimans, 2022). During training, this is done by simply adding to the sinusoidal timestep embeddings an additional learned class embedding from a set of $J + 1$ options $g_0, \dots g_J$. Here, option $g_0$ is used as the "null" class embedding applied when no additional contextual information is provided and trained with a given "class-dropout" probability. During inference, given access to a task label $j \in (1, .., J)$, we can

then construct a "guided" target prediction $\hat{x}_g$ for Eqn. 4:

$$\hat{x}_g = w_g \times f_\theta(x_t, t, g_j, c) - (1 - w_g) \times f_\theta(x_t, t, g_0, c), \quad (7)$$

where $w_g \geq 1$ is the guidance strength parameter. This method effectively provides diffusion models with targeted generation capabilities and plays a key role in their state-of-the-art computer vision performance (Dhariwal & Nichol, 2021). Moreover, it allows users to trade off general purpose with task-specific expertise, potentially allowing to overcome the impractical need for prompt engineering LMs.

## 4. Experimental Results

In this section, we provide descriptions for the implementation specifics, training, and evaluation of our new L2D method. Then, we present comprehensive quantitative results, evaluating the benefits of L2D across state-of-the-art LMs of different sizes from the Llama 3 (Dubey et al., 2024) and Qwen 2.5 families (Hui et al., 2024). Lastly, we focus on Llama 3.2 1B Instruct to study the properties of L2D in greater depth – showing its complementarity to traditional finetuning and search approaches, and also pushing performance with further advances from the diffusion literature, such as adaptive ODE solvers and classifier-free guidance.

To complement this section, we refer to Appendices A and B for a full set of hyper-parameters, further implementation details, and comprehensive descriptions of our datasets and tasks. Furthermore, we refer to Appendix C for thorough ablations of L2D and our baselines, together with Appendix D for results on additional benchmarks, analyses of additional extensions, detailed per-task performance tables, and comparisons with additional scaling approaches including the concurrent R1-style reasoning framework (Guo et al., 2025; Muennighoff et al., 2025).

### 4.1. Implementing, Training, and Evaluating L2D

As described in Section 2, our main L2D implementation adapts the frozen pre-trained model parameters with LoRA (Hu et al., 2021), efficiently reusing them in the diffusion path. Thanks to this design choice, training L2D is also relatively inexpensive and scalable, as the number of optimized parameters needed for backpropagation is dominated by the weights for the vocabulary of our new diffusion path, which do not grow with more layers. We employ $\sigma = 64$ for the standard deviation of the base distribution $p_0$, as the discrete nature of language makes token classification trivial for low noise levels and we want to regularize against the model's most influential diffusion steps being concentrated early on during inference. Similarly to related work (Dieleman et al., 2022; Gulrajani & Hashimoto, 2024), we employ a small diffusion dimension $\bar{d} = 256$ and rescale the inputs for $f_{\theta_d}$ such that the standard deviation of each component of $x_t$ has expectedly unit variance at

all timesteps $t$. While we did not consider it, we note that further decreasing $\bar{d} = 16$ could be an option to explore to make our method's optimized parameter count closer to LoRA and further reduce its cost, which has been shown viable by Dieleman et al. (2022). In all main results, we perform multi-step inference with a midpoint solver and 8 discretization levels, resulting in only 15 evaluations of $f_{\theta_d}$.

Typical applications of modern LMs involve processing a large fixed context of tokens before tackling the target task, such as user-provided prompts or fetched background resources. We note that this first step does not involve any active generation which could make use of improved reasoning skills. Thus, in contrast to prior diffusion LMs trained with unmasked pre-training language data, we fine-tune L2D on an instruction-following dataset targeted for tasks requiring non-trivial cognitive abilities, such as math and coding (Allal et al., 2024). As a consequence, L2D's learning signal is focused on powering the LM's conditional generation capabilities in complex problems – reflecting the conditions that would potentially benefit most from test-time scaling. We train each method for 1 epoch with the AdamW optimizer (Loshchilov, 2017), 100 warmup steps up to a tuned learning rate, and a linear decay afterward.

We evaluate L2D on challenging generation tasks broadly focused on math, coding, and general knowledge in a 5-shot setting. We choose to keep our evaluation consistent across all our tasks, without task-specific system prompts, sampling parameters, or involved answer extractions. Since L2D scaling does not provide a direct way for logits manipulation and due to the stochasticity requirements of pass@k evaluation for coding, we employ a simple untempered sampling strategy for generation. In Appendix D, we compare the effects of our evaluation setup with a close replication of the one from Dubey et al. (2024) on a sample task for all our main baselines, and provide further discussion about our scaling approach's current sampling constraints. We consider the following tasks: GSM8K (Cobbe et al., 2021) and competition MATH (Hendrycks et al., 2021b) to evaluate mathematical reasoning; HumanEval (Chen et al., 2021) and MBPP (Austin et al., 2021b) for coding skills; together with MMLU (Hendrycks et al., 2021a) and MMLU-Pro (Wang et al., 2024) to assess knowledge retention. However, due to its targeted design, we note that our training dataset is not meant to provide our models with new real-world knowledge that would be directly relevant to this last general knowledge category.

### 4.2. L2D Across Modern Large Language Models

In Table 1, we provide quantitative results after training L2D on top of four different LMs spanning different model families and scales. L2D yields consistent improvements that are particularly evident in the math and coding tasks,

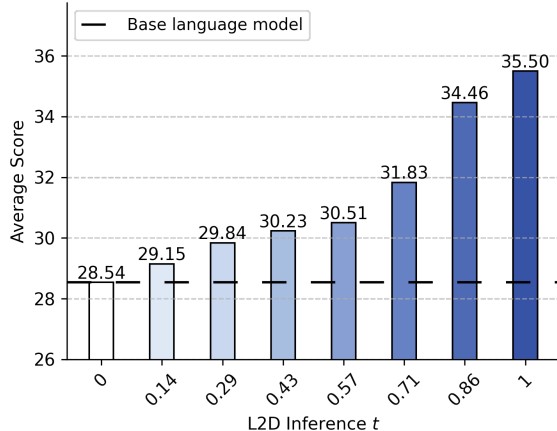

*Figure 3.* **Diffusion performance evolution.** Performance with the progression of the timestep $t$ within L2D's diffusion process.

the focus of our targeted training dataset, while optimizing a small fraction of the original weights (less than $6\%$ for Llama 1B and $3.5\%$ for Llama 8B). Although expectedly more limited, we still find some benefits in general knowledge tasks, indicating that the inductive bias from multi-step inference might also allow the model to better extract pre-acquired knowledge even beyond the finetuning corpus. Overall, we believe these results highlight the generality and effectiveness of L2D, allowing LMs to go beyond pure autoregression and harness some of the scaling properties of the diffusion framework, in line with this work's primary goal.

To disentangle the benefits of our method from our choice of data, we compare L2D with both LoRA and full weight finetuning baselines. As shown in our results, these traditional strategies appear to yield lower overall benefits with even frequent performance drops for the Llama instruct models on the coding problems. This is consistent across our models/task combination with the sole exception of the MBPP task with Qwen 7B, where the performance of L2D (76.79) comes at a close second behind the LoRA baseline (79.60). In Appendix D, we show that finetuning the base versions of Llama does not experience similar drops on coding tasks but fails to achieve competitive performance, suggesting that the private datasets employed in the instruction finetuning phases of these models might be superior to our public sources for certain problems. Nonetheless, L2D empirically shows consistent performance gains for all models, even in coding, indicating that its empirical properties are qualitatively different from traditional weight optimization: augmenting the model to leverage past computation and improve future predictions, without suffering the potential downsides of trying to alter its capabilities and knowledge.

### 4.3. Analysis and Extensions

**Inference-time diffusion scaling.** In Figure 1, we show the performance of L2D while simply scaling the number

*Table 1.* **Quantitative L2D evaluation** Performance and aggregated statistics for the considered math, coding, and general knowledge problems. All tasks are evaluated in a consistent 5-shot setting, and coding performance is measured under the pass@10 metric (Chen et al., 2021).

| Method/Task | Mathematics | | Coding | | General Knowledge | | Overall | |
|---|---|---|---|---|---|---|---|---|
| | GSM8K | MATH | HumanEval | MBPP | MMLU | MMLU-Pro | Average Score | Parameters |
| Llama 3.2 1B Instruct | 13.86 | 10.00 | 45.26 | 50.00 | 38.46 | 13.63 | 28.54 | - |
| + LoRA finetuning | 26.29 | 11.06 | 42.45 | 47.20 | 38.24 | 14.56 | 29.97 | 3M |
| + full finetuning | 33.48 | 12.40 | 32.08 | 30.00 | 39.57 | 14.70 | 27.04 | 1235M |
| + L2D (Ours) | **38.86** | **17.18** | **47.80** | **51.80** | **41.99** | **15.35** | **35.50** | 73M |
| Qwen 2.5 1.5B Instruct | 13.56 | 16.04 | 69.18 | 58.80 | 57.18 | 25.54 | 40.05 | - |
| + LoRA finetuning | 45.68 | 21.79 | 61.00 | 63.80 | 54.47 | 23.62 | 45.06 | 3M |
| + full finetuning | 50.45 | 23.34 | 65.41 | 51.80 | 52.63 | 22.59 | 44.37 | 1543M |
| + L2D (Ours) | **53.03** | **31.91** | **69.81** | **66.60** | **58.53** | **26.16** | **51.01** | 103M |
| Llama 3.1 8B Instruct | 51.97 | 23.05 | **83.65** | 70.20 | 63.83 | 31.85 | 54.09 | - |
| + LoRA finetuning | 69.70 | 27.21 | 78.62 | 70.40 | 60.37 | 29.38 | 55.95 | 13M |
| + full finetuning | 65.53 | 22.59 | 68.54 | 56.60 | 49.28 | 20.37 | 47.15 | 8030M |
| + L2D (Ours) | **75.61** | **35.69** | **83.65** | **71.03** | **66.69** | **35.28** | **61.33** | 281M |
| Qwen 2.5 7B Instruct | 5.61 | 18.34 | 87.42 | 58.60 | **71.41** | 38.51 | 46.65 | - |
| + LoRA finetuning | 70.08 | 33.82 | 88.05 | **79.60** | 69.39 | 39.12 | 63.34 | 10M |
| + full finetuning | 69.55 | 33.67 | 84.91 | 69.60 | 59.47 | 28.32 | 57.59 | 7615M |
| + L2D (Ours) | **82.80** | **43.62** | **91.20** | 76.79 | 71.11 | **39.96** | **67.58** | 233M |

of diffusion steps performed during inference. Moreover, in Figure 3, we show how performance varies within the L2D diffusion process as a function of $t$. In both cases, we expectedly observe a monotonic increase in overall LM performance, clearly analogous to the scaling properties of the diffusion framework for image modeling. Furthermore, comparing the scores of the highest and our default choice of 15 evaluations, in Figure 1 or Table 2, shows that over 90% of the performance boost can be retained without excessive overhead costs. These results evidence that the efficiency benefits of diffusion formulations based on rectified flows empirically transfer to the language domain, allowing effective generation in a handful of steps (Liu et al., 2022).

**Adaptive diffusion process.** In the first section of Table 2, we evaluate scaling compute using L2D with an adaptive second-order Runge-Kutta ODE solver (Fehlberg, 1969), running inference for 118.33 steps on average. Remarkably, this extension allows the Llama 1B model to exceed the highest previous results obtained with the midpoint solver and a fixed number of 127 steps – notably showing the effectiveness of adaptively tuning compute based on the diffusion errors for each generated token. In line with these observations, as illustrated in Figure 4, we find the number of steps to visibly vary between different tasks. For instance, when dealing with the challenging MATH and coding benchmarks (whose performance is provided in the pass@10 regime) the adaptive solver intuitively takes a larger number of steps than for GSM8K. Furthermore, we find that the tasks requir-

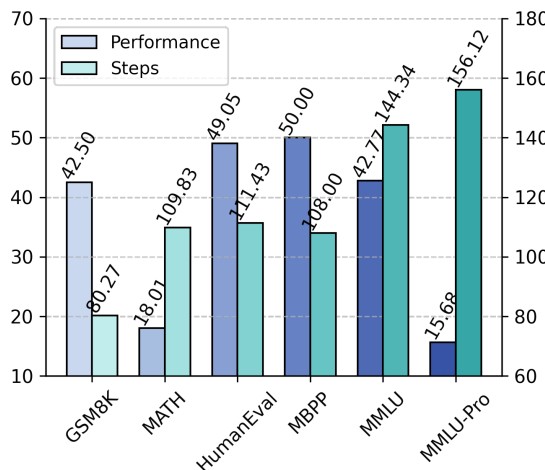

*Figure 4.* **Adaptive LM scaling** Performance (left) and average steps (right) across tasks using L2D with an adaptive ODE solver.

ing the model to provide an answer in a single token without allowing an initial reasoning trace (MMLU and MMLU-Pro) are distinctively the ones where the solver takes the most steps. These findings appear to suggest that integrating advanced solvers can provide L2D the ability to dynamically adapt compute to compensate for increasingly challenging settings and go beyond the current dependence of LMs on heuristic chain-of-thought traces (Wei et al., 2022).

**Full $f_{\theta_d}$ optimization and weight finetuning.** In the second section of Table 2, we show the effects of extending L2D with additional trained components. First, we examine

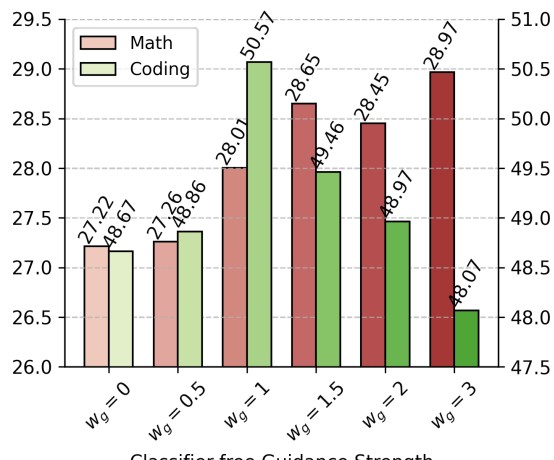

*Figure 5.* **Classifier-free guidance** Performance on the math (left) and coding tasks (right) varying L2D's guidance strength $w_g$.

*Table 2.* **L2D extensions.** Summarized performance statistics.

| Method/Metric | Math | Coding | All tasks | Params. |
|---|---|---|---|---|
| Llama 3.2 1B Instruct | 11.93 | 47.63 | 28.54 | - |
| + L2D | 28.02 | 49.80 | 35.50 | 73M |
| + L2D (127 steps) | 28.39 | **51.90** | 36.24 | 73M |
| + L2D (adaptive solver) | **30.26** | 49.53 | **36.34** | 73M |
| + L2D (full $f_{\theta_d}$ ft.) | 27.60 | **50.52** | 35.63 | 992M |
| + LoRA finetuning | 18.68 | 44.82 | 29.97 | 3M |
| + L2D (from LoRA ft.) | 29.19 | 48.45 | 35.51 | 76M |
| + full finetuning | 22.94 | 31.04 | 27.04 | 1235M |
| + L2D (from full ft.) | **33.37** | 43.37 | **35.84** | 1309M |
| + tuned token search | 27.76 | 49.35 | 33.83 | - |
| + L2D and token search | **35.95** | **49.79** | **38.57** | 73M |
| + L2D (guidance, $w_g = 1$) | 28.01 | **50.57** | 35.55 | 73M |
| + L2D (guidance, $w_g = 1.5$) | 28.65 | 49.46 | 35.62 | 73M |
| + L2D (guidance, tuned $w_g$) | **29.14** | **50.57** | **36.26** | 73M |

going beyond LoRA and optimizing the full set of parameters of $f_{\theta_d}$ (still initialized from the LM's frozen blocks). We find this simple change leads to improvements in L2D's overall performance, especially visible in the coding tasks. However, we note these benefits come with a non-negligible additional resource cost, a comparable trade-off to the one between traditional LoRA and full weight finetunings of LMs. Second, we study the effects of training L2D from already finetuned model checkpoints with these same traditional approaches. Our results confirm that L2D is fully compatible with direct parameter optimization, achieving some of our highest results on math where both methods were individually beneficial. Moreover, L2D also largely fills the performance drop observed when directly altering the weights of the Llama model on coding, further evidencing its synergy with traditional weight finetuning approaches.

**L2D and search.** In the third section of Table 2, we com-

pare and integrate L2D with traditional ways of increasing compute by searching over the space of generated tokens. In particular, using domain knowledge, we combine different effective heuristics to evaluate partial generations, such as the token sequence's likelihoods, lengths, and repetitions – which we tune by task category. We then use the resulting scores by performing a beam search over the generated sequences, keeping a set of 15 hypotheses to match the default number of L2D steps. Although the benefits of token search with the instruct model remarkably appear beyond traditional weight finetuning, even nearing the ones of L2D on coding, we note its cost and complexity are notably superior to our method: each L2D step only executes the far cheaper $f_{\theta_d}$, while each searched hypothesis even requires its own separate KV cache. Yet, by combining beam search with our method, each with half the original budget, we obtain the highest performance recorded by our extensions. We believe these results show how L2D makes diffusion highly complementary with traditional approaches for test-time scaling, and its future potential to accelerate progress toward advancing its current bounds (Liu et al., 2023; Brown et al., 2024; Snell et al., 2024).

**Classifier free guidance.** In the last section of Table 2, we illustrate the effects of integrating classifier-free guidance into L2D. As detailed in Appendix B, we partition the training data into the subsets most relevant for math, coding, and general knowledge to reflect the nature of the examined tasks. Then, by simply conditioning $f_{\theta_d}$ on the resulting labels during test time, our results demonstrate visible performance gains, further amplified by raising the guidance strength $w_g$. Yet, as shown in Figure 5, we find the optimal value for $w_g$ varies greatly across task categories, with single-answer math tasks benefiting from much higher guidance strengths than the pass@10 coding setting. This dichotomy mirrors the well-known trade-off between IS and FID metrics with traditional guided diffusion models (Salimans et al., 2016; Heusel et al., 2017). In fact, exploiting this property with per-domain tuning of $w_g$ even attains gains similar to running the unguided L2D for 127 steps with only 15. We believe these results further demonstrate the potential of the L2D framework to advance language modeling and bring to LMs some of the key advances that played a crucial role in establishing diffusion as state-of-the-art in computer vision (Dhariwal & Nichol, 2021).

## 5. Related Work

There have been several proposed generalizations of the diffusion process for discrete token spaces. Many works in this area focused on sequence-to-sequence tasks and multi-step generation (Reid et al., 2022; Zheng et al., 2023; Sahoo et al., 2024) by extending the seminal D3PM (Austin et al., 2021a). Other discretizations have seen success even for im-

age and biological data (Hoogeboom et al., 2021; Campbell et al., 2024). Of particular relevance, the recent SEDD (Lou et al., 2024) and discrete flow matching (Gat et al., 2024) demonstrated the early potential of this direction, making concrete strides in approaching small-scale traditional LMs.

Most related to our work, continuous diffusion LMs instead adapt the Gaussian diffusion framework to the language domain (Savinov et al., 2021; Li et al., 2022). This area has seen rapid evolution with techniques such as self-conditioning (Chen et al., 2022), new approaches to embed tokens in continuous spaces (Strudel et al., 2022; Mahabadi et al., 2023), and extensions to encoder-decoder domains (Yuan et al., 2022). In particular, CDCD (Dieleman et al., 2022) brought key advances also employed in this work, such as cross-entropy optimization and token normalization. Attempting to scale this line of work, PLAID (Gulrajani & Hashimoto, 2024) managed to train a 1B model outperforming a 124M GPT2 (Radford et al., 2019).

Similar in purpose but diverging from L2D's design, other works also aimed at combining the properties of LMs and diffusion. For instance, DiffusionBERT (He et al., 2022) proposed to use a pre-trained BERT model (Devlin, 2018) to accelerate masked diffusion training (Austin et al., 2021a). In addition, the SSD framework (Han et al., 2022; 2023) trained autoregressive and diffusion models together to act on different hierarchical language levels. Lastly, DGLM (Lovelace et al., 2024), proposed to learn a diffusion model on the latent space of an encoder-decoder LM to introduce classifier-free guidance support.

## 6. Discussion and Future Work

In this work, we provide concrete steps toward a new generation of autoregressively-trained LMs with the scaling capabilities of diffusion. We show how, after a small fine-tuning phase, L2D enables trading test-time compute for performance, providing higher and highly complementary benefits to further training and search-based optimizations. Additionally, we demonstrate how our new method provides LMs with the key properties of diffusion models, enabling effective adaptive computation and domain guidance expertise specific to user demands. However, the L2D framework still faces limitations left to be addressed by future work, as by scaling compute using a continuous diffusion LM (Dieleman et al., 2022) the model loses direct access to its ground-truth confidence scores, and a mechanism for simple logit manipulation – factors we further discuss and analyze in Appendix D. Concurrently with our work, scaling LMs with RL finetuning (Jaech et al., 2024; Guo et al., 2025) has emerged had another popular option, yet, still fully grounded in the space of tokens. To this end, combining the two approaches by harnessing RL training for L2D is another interesting future research direction, drawing inspiration from recent work in RL finetuning of diffusion models in computer vision (Black et al., 2023; Wallace et al., 2024). We hope this first work provides new inspiration for unifying the strengths of the foundational autoregressive and diffusion paradigms, which power some of the greatest milestones yet seen in AI.

## Acknowledgements

The authors would like to thank Stefano Peluchetti for providing important discussion and feedback to earlier versions of our work. Furthermore, we would like to thank the anonymous ICML reviewers and area chair for providing valuable feedback and suggestions to improve our work.

## Impact Statement

This paper presents work whose goal is to advance the field of Machine Learning. There are many potential societal consequences of our work, none which we feel must be specifically highlighted here.

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

*Table 3.* Implementation hyper-parameters of the weight finetuning baselines and L2D.

| Hyper-parameter | Weight finetuning | L2D |
|---|---|---|
| Flow hidden dimensionality $\bar{d}$ | – | 256 |
| Timestep embedding dimensionality | – | 256 |
| Diffusion path conditioning hidden dimensionality | – | 256 |
| Noise scaling ratio $\sigma$ | – | 64 |
| Optimizer | AdamW | AdamW |
| Warmup steps | 100 | 100 |
| Maximum learning rate | $1 \times 10^{-5}$ | $1 \times 10^{-4}$ |
| Final learning rate | $1 \times 10^{-6}$ | $1 \times 10^{-6}$ |
| Decay | Linear | Linear |
| LoRA alpha | 64 | 32 |
| Batch size | 32 | 32 |
| Training epochs | 1 | 1 |
| Maximum sequence length | 2048 | 2048 |
| Timestep training sampling $t$ | – | Uniform |
| ODE solver | – | Midpoint |
| Total diffusion budget $T$ | – | 15 |
| ODE velocity | – | Constant (Liu et al., 2022) |

## A. Implementation Details

### A.1. Language modeling hyper-parameters

We provide a full set of the default hyper-parameters for our baseline approaches and L2D in Table 3, including details about the training, inference, and modeling design of our new approach. In particular, we note that training is performed using the AdamW (Loshchilov, 2017) optimizer with a simple linear decay after a brief warmup phase, as we did not find any significant benefit from integrating more complex cosine schedules. As exemplified and detailed in Appendix C, we swept the learning rate and the other key hyper-parameters of each approach to ensure their efficacy. Our maximum sequence length, however, was selected for efficiency considerations from the quadratically scaling costs of transformer architectures, as we found monotonic performance improvements when increasing its value in preliminary experiments. For our beam search strategy, we score each partial completion based on the model's total loglikelihood divided by $L^{p_L}$, where $L$ is the current length and $p_L$ is a hyper-parameter to bias toward longer or shorter generations. We sample completions after each steps and, to further improve diversity, also at the end of the full procedure, by treating the resulting final scores as logits, which we divide by a per-task tuned temperature. We found this sampling approach particularly helpful for the pass@k coding class, which otherwise would be hurt by the lower resulting diversity.

### A.2. Inference L2D specifics

As described in Sections 2 and 3, we perform inference by starting the diffusion process from noise $x_0 \sim N(0, \sigma^2 I)$, and iteratively update $x_t$ at each diffusion step using predictions $\hat{x}$ sampled from the logits. We can perform this process with any ODE solver by discretizing the timestep interval $[0, 1]$ into a set of subintervals and integrating each segment with an $n$-th order approximation. In our default implementation, we integrate $[0, 1]$ with eight endpoints, i.e., at $S = \left(0, \frac{1}{7}, \frac{2}{7}, \frac{3}{7}, \frac{4}{7}, \frac{5}{7}, \frac{6}{7}, 1\right)$. Thus, with the second-order midpoint method, we perform two forward passes with the diffusion path to integrate each of the seven resulting subintervals $[S_i, S_{i+1}]$: once to compute the initial slope $dx_{S_i}$ at $t = S_i$ with diffusion token input $x_{S_i}$; and a second time at $t = S_i + h$ with diffusion token input $x_{S_i+h} = x_{S_i} + h dx_{S_i}$ yielding $dx_{S_i+h}$, where $h = \frac{S_{i+1}-S_i}{2}$. The output of the subinterval integration is then used to compute the value of its endpoint $x_{S_{i+1}} = x_{S_1} + 2h dx_{S_i+h}$, and the process is repeated for the next subinterval. One final forward pass is then done through the model to obtain and sample from the logits at the final step, resulting in a total diffusion budget of $T = 15$.

To avoid our proposed sampling procedure with higher-order solvers affecting the final diffusion token prediction by providing an embedding unseen from training averaged from different tokens, we found two implementation details useful

*Table 4.* Overview of evaluation datasets for the considered tasks and their characteristics.

| Dataset (subset) | Huggingface Repository | Split | Few-shot split | Size |
|---|---|---|---|---|
| InstructHumanEval | `codeparrot/instructhumaneval` | test | test | 159 |
| MBPP (full) | `google-research-datasets/mbpp` | test | prompt | 499 |
| GSM8K (main) | `openai/gsm8k` | test | train | 1,319 |
| MATH | `lighteval/MATH` | test | train | 4,347 |
| MMLU (all) | `cais/mmlu` | test | dev | 13,666 |
| MMLU-Pro | `TIGER-Lab/MMLU-Pro` | test | validation | 11,955 |
| PIQA | `ybisk/piqa` | validation | train | 1,838 |
| ARC-Easy | `allenai/ai2_arc` | test | validation | 2,376 |
| ARC-Challenge | `allenai/ai2_arc` | test | validation | 1,172 |

in practice. First, we linearly anneal the sampling temperature for the diffusion velocity toward zero with the progression of $t$ or simply take the most likely velocity. Second, we end the diffusion procedure slightly earlier at $t = 1 - \frac{1}{\sigma}$ as similarly done by Dieleman et al. (2022). However, we note that we did not find this last implementation detail always necessary with fixed-step first and second-order solvers, and only employed it for the adaptive and RK4 solvers (Fehlberg, 1969) part of our extensions evaluated in Section 4.3 and Appendix D. Furthermore, for the analyzed adaptive solver, we employed both absolute and relative thresholds to regulate the step size with values of $3 \times 10^{-4}$ each.

As explained in the main text our efficient design allows us to only compute the output of the model's pre-trained main path once during generation by simply storing it together with the KV cache. Then, by exploiting the fact that the main path is independent of the diffusion path until the final layer, we simply collect the updated residuals from the smaller-sized $f_{\theta_d}$ which take as input the latest diffusion token $x_t$ containing the compute and information gathered during all previous diffusion steps. Lastly, we want to acknowledge the `torchdiffeq` (Chen, 2018) library, which we use in our implementation to compute the diffusion path with L2D.

## B. Datasets

### B.1. Training Dataset Composition

Our targeted training and validation data used for L2D and our baselines is a carefully extracted combination of different subsets of the recent large open-source **SmolTalk** dataset (Allal et al., 2024). In particular, its specific composition was devised for the best performance with traditional weight finetuning approaches and for correlation to downstream reasoning tasks such as mathematics and coding. The adopted SmolTalk components include the subsets corresponding to `self-oss-instruct`, `metamathqa-50k`, `numina-cot-100k`, and `openhermes-100k`. Furthermore, we also extract and include a part of the examples from the `smol-magpie-ultra` subset – only considering data points with a category belonging to either `"coding"`, `"data-analysis"`, `"information-seeking"`, `"math"`, or `"reasoning"`. Lastly, we also note that we discard examples whose length exceeds 2048 tokens, matching the maximum considered sequence length employed during training. In total, the produced training and validation datasets contain 892,283 and 46,848 examples, respectively.

### B.2. Evaluation datasets

As described in Section 4 and in line with the training data, our evaluation suite comprises popular and challenging coding, math, and general knowledge tasks. Together with the sample from each of the tasks problems, we provide the model with a fixed 5-shot context from the task's data with either the first or equally spaced-out indexes (in case the task data is not i.i.d.) not included in the evaluation. We format the few-shot context as a past conversation adhering to the instruct LMs default tokenizers. In Table 4, we provide a summary of the data sources used for our evaluation, including for the additional tasks evaluated in Appendix D. We also provide high-level descriptions of our integrations and answer extraction procedures below:

**InstructHumanEval** is a coding dataset designed to assess instruction finetuned models. It extends the original HumanEval (Chen et al., 2021) and prepends each prompt with a natural language instruction that describes the coding problem.

The tasks typically involve writing Python functions that meet specific requirements. We compute *pass@1*, *pass@5*, and *pass@10* by executing model generations on provided unit tests.

**MBPP** (Multiple Basic Programming Problems, Austin et al. 2021b) contains programming problems written in natural language along with their solutions in Python. Following InCoder (Fried et al., 2023) and BigCode Evaluation Harness (Ben Allal et al., 2022), we include one unit test case in each prompt. Similarly, *pass@1*, *pass@5*, and *pass@10* are calculated by verifying model generations on unit tests.

**GSM8K** (Grade School Math 8K, Cobbe et al. 2021) is a dataset of grade school math word problems. Each problem requires breaking down the solution into several steps and applying basic arithmetic operations. A response has the format `"{multistep reasoning} ### {final answer}"`. We extract the final answer and compare it against the ground truth to compute exact match accuracy.

**MATH** (Mathematics Aptitude Test of Heuristics, Hendrycks et al. 2021b) consists of problems from mathematics competitions, including the AMC 10, AMC 12, AIME, and more. Each MATH response describes a full step-by-step solution and the final answer is wrapped in `\boxed{}`. We match and parse the content in `\boxed{}`, then compute accuracy by comparing it with the ground truth. In case, no `\boxed{}` answer is found, we simply take the final generated number as the model's response.

**MMLU** (Massive Multitask Language Understanding, Hendrycks et al. 2021a) is a broad evaluation benchmark testing knowledge across 57 different subjects, including humanities, STEM, social sciences, and more. The questions are in a multiple-choice format and require both general knowledge and specialized understanding. Options in a question are marked by letters from "A" to "D", and an answer is a single option letter. We report the accuracy of predicted option letters.

**MMLU-Pro** (Wang et al., 2024) presents more challenging multiple-choice questions that focus on professional knowledge. It extends 4 options in MMLU to 10 options (i.e. "A" to "J").

### B.3. Classifier-free Guidance Conditioning

As described in Sections 3 and 4, in our classifier-free guidance extension, L2D conditions on explicitly provided labels that reflect the nature of the examined tasks. Matching the task categories from our tables, we use the "math", "coding", and "general knowledge" labels to partition both the training and evaluation dataset for the considered tasks, as shown in Table 5. We believe that more fine-grained partitionings might allow L2D to develop even more nuanced capabilities. To this end, we believe our approach might have future untapped potential for the personalization of LMs, where different labels could provide the model contextual information to target behavior toward individual users through diffusion.

## C. Parameter Studies and Ablations

### C.1. Learning Rate

At the beginning of this work, we performed thorough LR sweeps for both L2D and the finetuning baselines on our training data. In practice, we found L2D benefits from much higher LR than direct weight finetuning, which we believe to be in line with our observation that traditional optimization can much more easily incur unwarranted knowledge loss than our new method. In Table 6, we provide summarized results locally modifying this parameter within $(1 \times 10^5, 3 \times 10^5, 1 \times 10^4)$. We note that going lower than $1 \times 10^5$ makes the performance of the finetuning baselines regress rapidly to the base model, defeating the very purpose of these approaches.

### C.2. Diffusion Schedule

As described in Sections 2 and 4, the choice of the standard deviation $\sigma$ for the base distribution $p_0$ is critical, implicitly defining the process that our diffusion-augmented LM will be learning. Too small or too large of a choice might concentrate the most relevant steps at either end of the diffusion interval, wasting both training and inference compute. In Table 7, we provide results with alternative values for $\sigma$ around our choice of $\sigma = 64$. As suggested by Dieleman et al. (2022), we note that the optimal diffusion schedule might evolve throughout training, with recent diffusion advances like time-warping being immediate directions for potential future improvements of our framework.

*Table 5.* Classifier-free guidance categories of the training and evaluation task datasets.

| Dataset | Category | Guidance Category |
|---|---|---|
| SmolTalk | `metamathqa-50k` | math |
| SmolTalk | `numina-cot-100k` | math |
| SmolTalk | `openhermes-100k` | general knowledge |
| SmolTalk | `self-oss-instruct/coding` | coding |
| SmolTalk | `self-oss-instruct/data-analysis` | general knowledge |
| SmolTalk | `self-oss-instruct/information-seeking` | general knowledge |
| SmolTalk | `self-oss-instruct/math` | math |
| SmolTalk | `self-oss-instruct/reasoning` | general knowledge |
| HumanEval | default | coding |
| MBPP | default | coding |
| GSM8K | default | math |
| MATH | default | math |
| MMLU | default | general knowledge |
| MMLU | `abstract_algebra` | math |
| MMLU | `college_mathematics` | math |
| MMLU | `elementary_mathematics` | math |
| MMLU | `high_school_mathematics` | math |
| MMLU | `high_school_statistics` | math |
| MMLU | `high_school_computer_science` | coding |
| MMLU-Pro | default | general knowledge |
| MMLU-Pro | `math` | math |
| PIQA | default | general knowledge |
| ARC-Easy | default | general knowledge |
| ARC-Challenge | default | general knowledge |

## C.3. Initialization

As detailed in Section 3, we initialize the weights of the diffusion path from the corresponding layers in the main path. The main goal behind this choice is to incentivize the model to learn a representation of the diffusion tokens close to one of the main path tokens and try to reuse the computation ability already present in the main path from pretraining. Our key hypothesis is that learning such a solution would be easier and provide a better inductive bias than learning the diffusion path from scratch. In Table 8, we provide this explicit ablation to validate our choice, showing a comparison with the full-finetuned version of L2D to equate the number of optimized parameters. However, we note that the performance of the randomly initialized L2D appears even lower than the less-costly LoRA version of our method – corroborating the usefulness and reusability of the parameters of open foundation models.

## C.4. Velocity Computation

As detailed in Section 2, to compute the target velocity, we simply sample $y_t$ from the output distribution of our L2D model $f_\theta(x_t, t, c)$. Then, we set $\hat{x} = V_{y_t}$. In contrast, Dieleman et al. (2022) opt to take the expectation over $f_\theta(x_t, t, c)$ directly as a weighted sum:

$$\hat{x} = \sum_y f_\theta(x_t, t, c)_y \times V_y. \qquad (8)$$

We provide results in Table 9, empirically comparing these choices. While in principle Dieleman et al. (2022)'s choice has the same expected value but lower variance than our sampling approach, we hypothesize the empirical advantage of our method when using deterministic ODE solvers comes from reinjecting some structured stochasticity, which Karras et al. (2022) showed might allow to better harness some of the self-correcting properties of the diffusion framework.

*Table 6.* Performance and aggregated statistics for different learning rates with L2D and traditional weight finetuning.

| Method/Metric | Mathematics | | Coding | | General knowledge | | Overall | |
|---|---|---|---|---|---|---|---|---|
| | GSM8K | MATH | HumanEval | MBPP | MMLU | MMLU-Pro | Average Score | Parameters |
| Llama 3.2 1B Instruct | 13.86 | 10.00 | 45.26 | 50.00 | 38.46 | 13.63 | 28.54 | - |
| + L2D (LR = $1 \times 10^{-5}$) | 38.41 | 17.39 | 44.65 | 43.25 | **43.47** | **15.57** | 33.79 | 73M |
| + L2D (LR = $3 \times 10^{-5}$) | **39.70** | **17.90** | 45.91 | 51.19 | 42.29 | 15.32 | 35.38 | 73M |
| + L2D (LR = $1 \times 10^{-4}$) | 38.86 | 17.18 | **47.80** | **51.80** | 41.99 | 15.35 | **35.50** | 73M |
| + full ft. (LR = $1 \times 10^{-5}$) | **33.48** | **12.40** | **32.08** | **30.00** | 39.57 | 14.70 | **27.04** | 1235M |
| + full ft. (LR = $3 \times 10^{-5}$) | 26.74 | 10.28 | 29.56 | 22.20 | 33.71 | 13.05 | 22.59 | 1235M |
| + full ft. (LR = $1 \times 10^{-4}$) | 15.91 | 7.54 | 20.75 | 7.40 | 25.91 | 11.37 | 14.81 | 1235M |

*Table 7.* Performance and aggregated statistics for L2D trained and evaluated with different standard deviation $\sigma$ of the base distribution $p_0 := N(0, \sigma^2 I)$.

| Method/Metric | Mathematics | | Coding | | General knowledge | | Overall | |
|---|---|---|---|---|---|---|---|---|
| | GSM8K | MATH | HumanEval | MBPP | MMLU | MMLU-Pro | Average Score | Parameters |
| Llama 3.2 1B Instruct | 13.86 | 10.00 | 45.26 | 50.00 | 38.46 | 13.63 | 28.54 | - |
| + L2D ($\sigma = 64$) | 38.86 | 17.18 | **47.80** | 51.80 | 41.99 | 15.35 | **35.50** | 73M |
| + L2D ($\sigma = 32$) | 37.50 | 16.82 | 45.28 | **52.38** | **42.22** | 15.54 | 34.96 | 73M |
| + L2D ($\sigma = 128$) | **41.06** | **18.45** | 44.03 | 46.83 | 42.09 | **16.06** | 34.75 | 73M |

# D. Extended Results

## D.1. Inference ODE Solvers

Our main experiments in Section 4 were collected with a second-order midpoint solver, an empirically robust choice in the traditional diffusion framework for different computational budgets (Lipman et al., 2024). When evaluating our framework with an adaptive solver, we also employed a second-order adaptive Runge-Kutta (RK) solver (Fehlberg, 1969). Here, we extend these results, analyzing additional fixed-sized solvers with different properties, to understand their behavior with L2D and our relatively small default diffusion budget. In Table 10, we provide results with the first-order Euler and fourth-order RK methods, evaluated for 15 and 17 steps (the lowest number that allows fourth-order integration above our default budget). In particular, we find that simpler solvers seem to work best, with Euler integration even slightly outperforming our midpoint method. These results appear consistent with the literature on fast diffusion methods (Liu et al., 2022). However, we note they might not necessarily hold for higher diffusion budgets as well (Karras et al., 2022).

## D.2. Timestep Schedules

For simplicity, in this work, we opted to sample timesteps $t \in [0, 1]$ uniformly during training. However, we note that there exist other choices recently developed that have been shown to provide empirical benefits for diffusions based on rectified flows (Esser et al., 2024). Thus, we validate the potential of these recent contributions for L2D and evaluate our method with the "cosmap" timestep schedule from Nichol & Dhariwal (2021). As shown in Table 11, this extension appears to yield consistent improvements over uniform sampling in all but one task, confirming how complementary advances from the diffusion literature can provide further improvements toward improving test-time LM scaling through our new framework.

## D.3. L2D Performance on Additional Tasks

In Table 12, we provide the performance of L2D and traditional weight finetuning strategies on additional evaluation settings and tasks from the language modeling literature. In particular, we report the pass@1 and pass@5 metrics for the HumanEval (Chen et al., 2021) and MBPP (Austin et al., 2021b) coding benchmarks, together with performance on the PIQA (Bisk et al., 2020), ARC-Easy, and ARC-Challenge (Clark et al., 2018) question-answering tasks. We note that these last three tasks are less relevant than the ones considered in Section 4, given our data curation strategy targeted toward math

*Table 8.* Performance and aggregated statistics for L2D ablating our reuse of the main pretrained path's weights $\theta_l$ to initialize the weights of the diffusion path $\theta_d$.

| Method/Metric | Mathematics | | Coding | | General knowledge | | Overall | |
|---|---|---|---|---|---|---|---|---|
| | GSM8K | MATH | HumanEval | MBPP | MMLU | MMLU-Pro | Average Score | Parameters |
| Llama 3.2 1B Instruct | 13.86 | 10.00 | 45.26 | 50.00 | 38.46 | 13.63 | 28.54 | - |
| + L2D (full $f_{\theta_d}$ ft.) | 37.50 | **17.71** | **49.05** | **52.00** | **41.98** | **15.52** | **35.63** | 992M |
| + L2D (full $f_{\theta_d}$ ft. from scratch) | **38.03** | 17.46 | 42.77 | 51.19 | 41.71 | 14.71 | 34.31 | 992M |

*Table 9.* Performance and aggregated statistics for L2D evaluated with the $\hat{x}$ estimate proposed by Dieleman et al. (2022) to compute the velocity.

| Method/Metric | Mathematics | | Coding | | General knowledge | | Overall | |
|---|---|---|---|---|---|---|---|---|
| | GSM8K | MATH | HumanEval | MBPP | MMLU | MMLU-Pro | Average Score | Parameters |
| Llama 3.2 1B Instruct | 13.86 | 10.00 | 45.26 | 50.00 | 38.46 | 13.63 | 28.54 | - |
| + L2D | **38.86** | 17.18 | **47.80** | **51.80** | **41.99** | **15.35** | **35.50** | 73M |
| + L2D (velocity from expectation) | 37.12 | **18.33** | 46.23 | 51.60 | 41.31 | 14.96 | 34.92 | 73M |

and coding problems. Remarkably, however, while weight finetuning appears to deteriorate performance across several model-task combinations, L2D once again provides much more consistent benefits throughout. These results are in line with our observations in the main text that by focusing on augmenting rather than altering the original model, L2D does not seem to suffer the potential pitfalls of traditional weight finetuning atop powerful instruct models.

### D.4. L2D and Best-of-N Scaling

In Section 4, we evaluate two approaches for "best-of-N" scaling. First, the token search baseline is itself an advanced version of best-of-N scaling, where the tuned beam-search scores are used as a heuristic metric to assess which is the best response. Second, we also consider best-of-N using ground-truth correctness, which assumes access to an oracle verifier and is typically only considered for coding, where the oracle could come in the form of a compiler and a set of test cases to solve. In fact, this is precisely what the pass@K metric used for Humaneval/MBPP considers. In Table 13, we also provide the pass@K performance of L2D and traditional weight on the remaining set of math and general knowledge tasks using the Llama 1B model, which could be viewed as an upper bound for any critic-based inference-scaling approaches. As shown, with access to an oracle verifier, simple repeated sampling is likely the preferred scaling approach. However, consistently with prior results, L2D is expectedly complementary to this scaling approach and remains an effective, viable strategy to push performance beyond best-of-N's inevitable saturation.

### D.5. L2D Performance with Base Models

As exemplified for the coding tasks in Section 4 and further evidenced in the above subsection, some of the private data involved in the instruction-tuning phases of state-of-the-art models seems to be more effective than publicly available sources. However, to validate our curated reasoning dataset, we trained and evaluated both our weight finetuning baselines starting from the base Llama 3.2 1B model. As shown in Table 14, without previous instruction tuning, both strategies seem to provide remarkable benefits across all considered tasks, with full weight finetuning achieving the highest overall scores, in clear contrast to the results atop the Llama 3.2 1B Instruct model.

### D.6. Current Sampling Constraints and Evaluation Differences

As described in Section 6, the new effective test time scaling framework of L2D still faces limitations, in terms of its flexibility and interpretability, left to be addressed by future work. In particular, scaling compute using a continuous diffusion LM (Dieleman et al., 2022) always introduces stochasticity into generation and makes the model's log likelihood directly dependent on the diffusion timestep $t$. Thus, this makes LMs evaluated with multi-step L2D scaling lose direct access to their ground-truth confidence scores and removes their inherent mechanisms for simple logit manipulation. However, unlike

*Table 10.* Performance and aggregated statistics for L2D evaluated with fixed-step ODE solvers of different order.

| Method/Metric | Mathematics | | Coding | | General knowledge | | Overall | |
|---|---|---|---|---|---|---|---|---|
| | GSM8K | MATH | HumanEval | MBPP | MMLU | MMLU-Pro | Average Score | Parameters |
| Llama 3.2 1B Instruct | 13.86 | 10.00 | 45.26 | 50.00 | 38.46 | 13.63 | 28.54 | - |
| + L2D (midpoint, 15 steps) | 38.86 | 17.18 | 47.80 | **51.80** | 41.99 | **15.35** | 35.50 | 73M |
| + L2D (Euler, 15 steps) | **39.77** | **17.30** | **48.42** | 50.20 | 42.19 | 15.30 | **35.53** | 73M |
| + L2D (RK4, 17 steps) | 39.70 | 17.28 | 45.91 | 51.19 | **42.41** | 14.95 | 35.24 | 73M |

*Table 11.* Performance and aggregated statistics for L2D trained with the sampling schedule from (Nichol & Dhariwal, 2021) for the diffusion timestep $t$.

| Method/Metric | Mathematics | | Coding | | General knowledge | | Overall | |
|---|---|---|---|---|---|---|---|---|
| | GSM8K | MATH | HumanEval | MBPP | MMLU | MMLU-Pro | Average Score | Parameters |
| Llama 3.2 1B Instruct | 13.86 | 10.00 | 45.26 | 50.00 | 38.46 | 13.63 | 28.54 | - |
| + L2D | 38.86 | 17.18 | 47.80 | **51.80** | 41.99 | 15.35 | 35.50 | 73M |
| + L2D (cosmap schedule) | **39.92** | **18.38** | **48.43** | 51.60 | **42.06** | **15.36** | **35.96** | 73M |

previous work trying to learn language diffusion models from scratch (Li et al., 2022; Dieleman et al., 2022; Gulrajani & Hashimoto, 2024), we note that when requiring access to the model's probabilities or for tasks that might benefit on particular sampling strategies, L2D's design always offers the possibility of running the model for a single step fully preserving the original capabilities and behavior of autoregressive LMs.

As detailed in Section 3, we choose to keep our evaluation consistent across all our tasks, without any task-specific system prompts, sampling parameters, or involved answer extractions. In Table 15, we compare the effects of our evaluation setup with a close replication of the one from Dubey et al. (2024) on the GSM8K task for all our main baselines. In particular, this implementation uses a much relaxed answer extraction that first looks for specific answer patterns even beyond the chain-of-thought examples, and if no answer is properly formatted, it still attempts to match the solution with any numerical value present in the response. Furthermore, this implementation also uses eight particular chain-of-thought examples and a greedy sampling generation strategy, even though L2D's behavior is not affected by logit manipulations. As shown in our results, all baselines appear to consistently benefit from our considered constraint relaxations with the additional chain-of-thought samples and the different generation strategy, especially the base Llama models, whose performance becomes very close to the one reported by Dubey et al. (2024) on this task implementation. We note that, on the small 1B models, also LoRA and full finetuning become significantly more effective, and only by combining them with L2D we were able to improve upon their performance for Llama. However, on the larger 8B Llama model, we find their inclusion comes at a detriment to the original model's performance, while L2D scaling leaves it relatively close to the original. Overall, we believe these results highlight once again that L2D should be viewed not just to replace but also to be used in combination with traditional weight finetuning, bringing to traditional LMs new compounding benefits that can be enabled on demand with extra compute.

### D.7. Reasoning and Other Test Time Scaling Approaches

In Table 16, we evaluate L2D with the concurrent RL-induced LM reasoning framework for test time scaling (Jaech et al., 2024; Guo et al., 2025). Since RL training requires expensive multi-node settings, far beyond L2D training, and appears mainly effective on very large LMs, we added results with the pre-trained DeepSeek R1 Distill Qwen 1.5B reasoning model (Guo et al., 2025). We used this model both as an additional baseline and as an extra base model from which to train L2D. As highlighted in our results, since the DeepSeek R1 model is trained on a recent private dataset, heavily focused on math, we find its performance exceeds the original Qwen 1.5B Instruct model on this task category. However, we find this comes at an expected actual loss in performance on coding and general knowledge, which our L2D approach avoids. Moreover, further fine-tuning this baseline with L2D achieves the highest results on math, even surpassing the much larger 7B and 8B non-RL models – as well as recovering a large part of the performance loss on the other tasks. In line with the

*Table 12.* Performance and aggregated statistics for L2D and our main ablations across all Llama and Qwen models for additional pass@k settings and tasks.

| Method/Task | Coding extended results | | | | Additional tasks | | | Overall |
|---|---|---|---|---|---|---|---|---|
| | HumanEval@5 | HumanEval@1 | MBPP@5 | MBPP@1 | ARC-Easy | ARC-Challenge | PIQA | Parameters |
| Llama 3.2 1B Instruct | 38.54 | 20.94 | 43.88 | 23.16 | 63.68 | 44.20 | 55.98 | - |
| + LoRA finetuning | 33.03 | 16.64 | 40.24 | 20.28 | 64.56 | 45.48 | 57.56 | 3M |
| + full finetuning | 27.47 | 16.42 | 22.39 | 7.66 | 67.59 | 43.60 | **58.11** | 1235M |
| + L2D (Ours) | **41.14** | **25.09** | **45.83** | **28.40** | **67.68** | **47.95** | 56.03 | 73M |
| Qwen 2.5 1.5B Instruct | 59.42 | 32.58 | 50.41 | 25.66 | 89.23 | 75.09 | 76.44 | - |
| + LoRA finetuning | 54.13 | 30.60 | 54.25 | 29.14 | 86.20 | 70.99 | 74.43 | 3M |
| + full finetuning | 55.63 | 30.50 | 42.50 | 17.72 | 86.70 | 70.90 | 74.05 | 1543M |
| + L2D (Ours) | **62.41** | **39.21** | **59.99** | **38.40** | **89.60** | **75.68** | **76.79** | 103M |
| Llama 3.1 8B Instruct | **78.32** | **55.47** | 65.04 | 47.70 | 92.59 | 80.20 | 81.23 | - |
| + LoRA finetuning | 71.66 | 44.37 | 64.08 | 41.22 | 90.61 | 78.84 | 78.94 | 13M |
| + full finetuning | 60.00 | 33.24 | 48.81 | 25.00 | 81.57 | 67.41 | 71.49 | 8030M |
| + L2D (Ours) | 77.10 | 53.96 | **66.08** | **48.12** | **92.97** | **82.85** | **83.64** | 281M |
| Qwen 2.5 7B Instruct | 83.83 | 67.30 | 54.60 | 39.88 | 96.04 | **89.59** | 86.51 | - |
| + LoRA finetuning | 81.17 | 53.02 | **72.76** | 47.42 | 95.33 | 87.63 | 85.31 | 10M |
| + full finetuning | 77.22 | 49.28 | 61.75 | 33.52 | 91.88 | 80.80 | 77.75 | 7615M |
| + L2D (Ours) | **86.27** | **68.58** | 70.53 | **48.43** | 96.04 | 88.65 | **86.74** | 233M |

other results, we believe these findings confirm that our new method should be viewed as potentially complementary to this recent reasoning framework. However, we note that evaluating these reasoning models distilled from RL was over 10x more expensive than vanilla L2D and did not work out-of-the-box, requiring us to modify the prompts and relax the answer extraction code for compatibility with "<think>/<answer>" style responses. Furthermore, while this has not been explored in this work, scaling training data and its quality, or even including an additional RL phase, could potentially allow L2D to achieve similar "latent" reasoning abilities on its own, and remains an open research direction.

In Table 17, we also compare L2D with higher quality chain-of-thought scaling (CoT) (Wei et al., 2022). For this comparison, we made versions of our tasks with new chain-of-thought few-shot examples designed to elicit better and longer reasoning. In particular, these examples were obtained by prompting Claude Sonnet 3.7 to provide more effective and longer chain-of-thought based on the heuristics recommended in Wang et al. (2022). We note this change significantly increased inference time, especially for our multiple-choice tasks, going from the models generating a single letter answer directly to producing lengthy reasoning traces beforehand (averaging 84 new tokens). As shown by our results, this tuned chain-of-thought prompting strategy indeed achieves improvements for both the base Llama model and our other finetuning baselines, albeit lower than our previous baseline results and L2D. Furthermore, in line with our other findings, using L2D models together with chain-of-thought prompting yields compounding test-time benefits, which we believe again evidences the synergy between our method and orthogonal scaling approaches that work by increasing generation length.

### D.8. Full L2D Extensions Results

In Tables 18 and 19, we provide the full set of results for the extensions to L2D analyzed in Section 4. As discussed in the main text, we find the effects of adaptive solvers and test-time advances like classifier-free guidance to be of remarkable importance, considerably beyond simply scaling the number of training parameters. We find these results quite analogous to similar findings from the diffusion literature (Karras et al., 2022), showing how L2D has the potential to open doors beyond the current language modeling framework, where data and training compute are the current predominant approaches for scaling.

*Table 13.* Performance of L2D and our main baselines for additional pass@K settings in Math and General Knowledge tasks, to provide a strict performance upper bound for any best-of-N approach.

| Method/Metric | Mathematics | | Coding | |
|---|---|---|---|---|
| | GSM8K | MATH | MMLU | MMLU-Pro |
| Llama 3.2 1B Instruct pass@1 | 13.86 | 10.00 | 38.46 | 13.63 |
| + LoRA finetuning pass@1 | 26.29 | 11.06 | 38.24 | 14.56 |
| + full finetuning pass@1 | 33.48 | 12.40 | 39.57 | 14.70 |
| + L2D pass@1 | **38.86** | **17.18** | **41.99** | **15.35** |
| Llama 3.2 1B Instruct pass@5 | 40.83 | 27.96 | 75.24 | 42.97 |
| + LoRA finetuning pass@5 | 58.64 | 30.68 | 73.60 | 43.39 |
| + full finetuning pass@5 | 63.33 | 32.35 | 76.59 | 42.94 |
| + L2D pass@5 | **67.35** | **40.05** | **77.34** | **43.74** |
| Llama 3.2 1B Instruct pass@10 | 52.65 | 37.20 | 87.87 | 60.80 |
| + LoRA finetuning pass@10 | 69.47 | 41.64 | 84.38 | 61.16 |
| + full finetuning pass@10 | 72.58 | 42.67 | 89.07 | 58.66 |
| + L2D pass@10 | **75.68** | **49.93** | **89.87** | **62.05** |

*Table 14.* Performance and aggregated statistics for the LoRA (Hu et al., 2021) and full weight finetuning baselines across both instruct and non-instruct versions of the Llama 3.2 1B LM.

| Method/Task | Mathematics | | Coding | | General knowledge | | Overall | |
|---|---|---|---|---|---|---|---|---|
| | GSM8K | MATH | HumanEval | MBPP | MMLU | MMLU-Pro | Average Score | Parameters |
| Llama 3.2 1B Instruct | 13.86 | 10.00 | **45.26** | **50.00** | 38.46 | 13.63 | 28.54 | - |
| + LoRA finetuning | 26.29 | 11.06 | 42.45 | 47.20 | 38.24 | 14.56 | **29.97** | 10M |
| + full finetuning | **33.48** | **12.40** | 32.08 | 30.00 | **39.57** | **14.70** | 27.04 | 7615M |
| Llama 3.2 1B | 2.05 | 2.10 | 16.98 | 11.60 | 26.51 | 11.20 | 11.74 | - |
| + LoRA finetuning | 4.55 | 2.53 | 22.64 | **28.80** | 25.39 | 11.42 | 15.89 | 10M |
| + full finetuning | **17.42** | **5.68** | **23.75** | 12.80 | **28.62** | **11.74** | **16.67** | 7615M |

*Table 15.* Performance of L2D and our main baselines on a relaxed GSM8K (Cobbe et al., 2021) task implementation, with a more permissive answer extraction, eight chain-of-thought prompts, and greedy sampling, matching the task implementation from Dubey et al. (2024).

| Method/Task | Mathematics | | Parameters |
|---|---|---|---|
| | GSM8K | GSM8K (greedy/relaxed) | |
| Llama 3.2 1B Instruct | 13.86 | 39.24 | - |
| + LoRA finetuning | 26.29 | 41.24 | 3M |
| + full finetuning | 33.48 | 46.36 | 1235M |
| + L2D | 38.86 | 43.41 | 73M |
| + L2D (from full ft.) | **46.89** | **48.11** | 1308M |
| Qwen 2.5 1.5B Instruct | 13.56 | 31.06 | - |
| + LoRA finetuning | 45.68 | 62.66 | 3M |
| + full finetuning | 50.45 | **63.18** | 1543M |
| + L2D | **53.03** | 59.85 | 103M |
| Llama 3.1 8B Instruct | 51.97 | **81.21** | - |
| + LoRA finetuning | 69.70 | 77.73 | 13M |
| + full finetuning | 65.53 | 75.53 | 8030M |
| + L2D | **75.61** | 80.98 | 281M |
| Qwen 2.5 7B Instruct | 5.61 | 47.42 | - |
| + LoRA finetuning | 70.08 | 80.53 | 10M |
| + full finetuning | 69.55 | 81.59 | 7615M |
| + L2D | **82.80** | **82.58** | 233M |

*Table 16.* **L2D comparison and integration with R1-style RL.** Summarized performance statistics. *Indicates modified task prompts, answer extraction, and evaluation compatible with <think>/<answer> style responses.*

| Method/Metric | Mathematics | Other tasks | All tasks | Parameters |
|---|---|---|---|---|
| Qwen 2.5 1.5B Instruct | 14.80 | 52.67 | 40.05 | - |
| + L2D | **42.47** | **55.28** | **51.01** | 103M |
| DeepSeek-R1-Distill-Qwen-1.5B* | 66.33 | 23.66 | 37.88 | 1543M |
| + L2D | **69.35** | **28.73** | **42.27** | 1647M |

*Table 17.* **L2D comparison and integration with chain-of-thought scaling.** Summarized performance statistics.

| Method/Metric | Mathematics | Coding | All tasks | Paramseters |
|---|---|---|---|---|
| Llama 3.2 1B Instruct | 11.93 | 47.63 | 28.54 | - |
| + LoRA finetuning | 18.68 | 44.82 | 29.97 | 3M |
| + full finetuning | 22.94 | 31.04 | 27.04 | 1235M |
| + L2D | **28.02** | **49.80** | **35.50** | 73M |
| + CoT | 13.97 | 48.81 | 29.64 | - |
| + LoRA finetuning and CoT | 19.35 | 46.84 | 30.94 | 3M |
| + full finetuning and CoT | 23.97 | 34.27 | 28.48 | 1235M |
| + L2D and CoT | **29.04** | **50.29** | **36.00** | 73M |

*Table 18.* Full per-task performance and aggregated statistics for the L2D extensions from Section 4.

| Method/Metric | Mathematics | | Coding | | General knowledge | | Overall | |
|---|---|---|---|---|---|---|---|---|
| | GSM8K | MATH | HumanEval | MBPP | MMLU | MMLU-Pro | Average Score | Parameters |
| Llama 3.2 1B Instruct | 13.86 | 10.00 | 45.26 | 50.00 | 38.46 | 13.63 | 28.54 | - |
| + L2D | 38.86 | 17.18 | 47.80 | **51.80** | 41.99 | 15.35 | 35.50 | 73M |
| + L2D (127 steps) | 38.86 | 17.92 | **52.20** | 51.60 | 41.87 | 14.96 | 36.24 | 73M |
| + L2D (adaptive solver) | **42.50** | **18.01** | 49.05 | 50.00 | **42.77** | **15.68** | **36.34** | 73M |
| + L2D (full $f_{\theta_d}$ ft.) | 37.50 | 17.71 | **49.05** | 52.00 | 41.98 | 15.52 | 35.63 | 992M |
| + LoRA finetuning | 26.29 | 11.06 | 42.45 | 47.20 | 38.24 | 14.56 | 29.97 | 3M |
| + L2D (from LoRA ft.) | 40.15 | 18.24 | 45.91 | 51.00 | 42.79 | 14.98 | 35.51 | 76M |
| + full finetuning | 33.48 | 12.40 | 32.08 | 30.00 | 39.57 | 14.70 | 27.04 | 1235M |
| + L2D (from full ft.) | **46.89** | **19.85** | 43.33 | 43.40 | **44.34** | **17.23** | **35.84** | 1309M |
| + token search | 36.44 | 19.07 | **48.91** | 49.80 | 35.33 | 13.44 | 33.83 | - |
| + L2D and token search | **46.21** | **25.69** | 47.80 | **51.79** | **43.29** | **16.65** | **38.57** | 73M |
| + L2D (guidance, $w_g = 1$) | 38.26 | 17.76 | **49.54** | **51.60** | 41.31 | 14.85 | 35.55 | 73M |
| + L2D (guidance, $w_g = 1.5$) | 39.24 | **18.06** | 47.73 | 51.19 | 42.17 | 15.35 | 35.62 | 73M |
| + L2D (guidance, tuned $w_g$) | **40.23** | **18.06** | **49.54** | **51.60** | **42.52** | **15.62** | **36.26** | 73M |

*Table 19.* Full per-task performance and aggregated statistics for L2D classifier-free guidance extension from Section 4 evaluated with different classifier strengths $w_g$.

| Method/Metric | Mathematics | | Coding | | General knowledge | | Overall | |
|---|---|---|---|---|---|---|---|---|
| | GSM8K | MATH | HumanEval | MBPP | MMLU | MMLU-Pro | Average Score | Parameters |
| $w_g = 0$ | 37.20 | 17.23 | 46.54 | 50.79 | 40.94 | 14.71 | 34.57 | 73M |
| $w_g = 0.5$ | 36.89 | 17.62 | 46.54 | 51.19 | 41.06 | 14.70 | 34.67 | 73M |
| $w_g = 1$ | 38.26 | 17.76 | **49.54** | **51.60** | 41.31 | 14.85 | 35.55 | 73M |
| $w_g = 1.5$ | 39.24 | **18.06** | 47.73 | 51.19 | 42.17 | 15.35 | **35.62** | 73M |
| $w_g = 2$ | 38.86 | 18.04 | 47.73 | 50.20 | 42.32 | 15.32 | 35.41 | 73M |
| $w_g = 3$ | **40.23** | 17.71 | 46.54 | 49.60 | **42.52** | **15.62** | 35.37 | 73M |

