# OpenReview forum: "Large Language Models to Diffusion Finetuning"
_ICML.cc/2025/Conference — ICML 2025 poster_

### Official Review · Reviewer_hLPT · 2025-03-10

**Overall Recommendation:** 3

**Summary:**

This paper proposes fine-tuning pretrained large language models (LLMs) using diffusion models to enable scalable test-time computation. By framing LLMs as single-step diffusions and introducing a small fraction of new parameters, the approach enhances multi-step reasoning, allows adaptive computational scaling, and incorporates guidance techniques while preserving the original model’s efficiency.

**Claims And Evidence:**

**1**. *Claim 1*:\
"We show that L2D significantly improves four different LMs on math, coding, and a variety of reasoning tasks; and that its benefits can be both superior and complementary to traditional finetuning and search." \
*Evidence 1*:\
From the experimental results, L2D achieves approximately 5 more score on average than LoRA, but at the cost of more than 20 times number of parameters. For tasks such as coding on MBPP and general knowledge, the enhancements seem to be marginal. Finetuning on Qwen 2.5 7B Instruct for MBPP even demonstrates that LoRA outperforms L2D. It leaves the question on the effectiveness and scalability of L2D.\

**Essential References Not Discussed:**

No, I think all the key references are included and discussed.

**Experimental Designs Or Analyses:**

Yes, the experimental design and analysis is sound for me.

**Methods And Evaluation Criteria:**

The proposed L2D is the compression of large language models achieved via fine-tuning for classifier-guided diffusion models. Because of the current lack of diffusion-based methods on language generation, it is natural to consider using diffusion models which can progressively generate contents, to predict next tokens. Since tokens are results generated from LLM, it is logically linear for me to fine-tune a diffusion path to compress LLM.

The evaluation criteria: scores, parameter sizes, strengths of guidance and the number of inference steps, are aligned with the evaluation of the compression performance mainly on efficiency and effectiveness. The benchmark datasets are the commonly used datasets to evaluate the performance of LLM and the compression for LLM.

Overall, the proposed methods and evaluation criteria make sense for me.

**Other Comments Or Suggestions:**

NA

**Other Strengths And Weaknesses:**

**Weaknesses**:\
*a. Clarity*: This paper definitely requires to be polished on writing, especially the structure of the sentences and the phrasing. It is very confusing to have consecutive multiple preposition phrases or clauses to describe one object, as it is easy for the audience to be lost in these descriptions and forget the object.\

*b. Quality*: The insights from the experimental results should be clearly articulated and discussed, for example, the trade-off between scores and the number of parameters. Some preliminary theoretical insights or propositions will be beneficial.

**Questions For Authors:**

**1**. How are the subset of the layers of LM being selected to be the building blocks of diffusion path? Are they randomly selected or hand-picked? Is there any special requirements of the selection to guarantee the performance with intuitive or theoretical insights? Is it the main technique to achieve the scalability?

**Relation To Broader Scientific Literature:**

This work fills the research gap in diffusion models for language generation. It is novel to compress LLM via a diffusion path. It opens the possibility to bridge diffusion models which is ubiquitous for visual content generation, with LLM which is designed for language generation. The experimental results in Figure 3 clearly demonstrate the power of diffusion paths as compressions of LLMs can even achieve better performance than LLMs in language generation after a relatively small number of inference steps.

**Theoretical Claims:**

There is no theoretical claim in this paper.

---

> ### Author Rebuttal · Authors · 2025-03-31
>
> We would like to thank the reviewer for their feedback and the time they dedicated to our review.
>
> **Claims and Evidence**
>
> In our [Table 1](https://anonymous.4open.science/r/rebuttal_l2d-4B0B/table1.png) results, across all 24 task/model combinations examined, our full weight finetuning and LoRA baselines improve performance by 1.7 and 6.23 on average over the base model. In contrast, L2D improves average performance by 11.52 (85% higher improvement than LoRA) and outperforms both finetuning baselines in 23/24 settings.
>
> As the reviewer correctly points out, for one case, in the MBPP task with Qwen 7B, the performance of L2D (76.79) is only second best, slightly behind the LoRA baseline (79.60). While this is not the case for any other task, or even any other model other than Qwen 7B on the same task, we believe this could be an indication that L2D should be viewed not just to replace but also to be used in combination with traditional weight finetuning. The potential of combining these approaches is also supported by our extensions results in the second section of [Table 2](https://anonymous.4open.science/r/rebuttal_l2d-4B0B/table2.png), where we show that L2D can be effectively used after full and lora finetunings of the base model with compounding benefits.
>
> Following the reviewer’s feedback, in our latest revision of this work, we modified Section 1 to be more precise with our wordings and avoid general statements (such as Claim 1) and extended Section 4 to specifically address the MBPP results as detailed above.
>
> Regarding the scalability of L2D, as the reviewer also pointed out, the additional parameters of our method as compared to the LoRA baselines mainly come from storing the weights for the vocabulary of our new diffusion LM path (the module denoted "MLP" [at the bottom of our architecture diagram](https://anonymous.4open.science/r/rebuttal_l2d-4B0B/l2d_architecture_diagram.png)). We would like to note that the number of these parameters does not grow with model size, making our method increasingly more efficient in relative terms with larger LMs. Nonetheless, even in for the LMs considered in this work, the total number of optimized parameters is still only a small fraction of the original model’s weight (less than 6% for Llama 1B and 3.5% for Llama 8B), with L2D being over an order of magnitude more parameter efficient than full finetuning.
>
> Following the reviewer’s feedback, we added the above discussion about parameter efficiency to the latest revision of our work to provide a better context of our method’s parameter efficiency and scalability. Furthermore, we also mention that decreasing the diffusion space dimension (e.g., from 256 used in our work to 16) could be an option to explore in future work to make our method’s parameter count closer to LoRA, which has been shown viable by some of the diffusion language models learned from scratch such as [1].
>
> [1] Continuous diffusion for categorical data, 2022.
>
> **Clarity**
>
> Following the reviewer’s feedback, we tried to identify in the text multiple instances where we used “*multiple preposition phrases or clauses to describe one object*” and tried to improve clarity by breaking them up and referring back to the object explicitly. For instance, we rewrote the sentence at the start of Section 2.1 to:
>
> “Gaussian diffusion decomposes the problem of generating new samples from a target unknown distribution p* from a source distribution q = N(0, I) over multiple 'simpler' steps. The Gaussian diffusion decomposition effectively reuses the intermediate information computed in the model's attempts in each previous step.”
>
> If the reviewer has any other specific example where text clarity could be improved, we hope they will not hesitate to point it out.
>
> **Questions**
>
> The building blocks of the diffusion path are constructed from all layers in the MLP modules and only the layers of the pre-trained self-attention modules that are used to compute the queries (used in the diffusion path for cross-attention), as detailed in the second paragraph of Section 3.1.
>
> Following the reviewer’s question, we added this detail more explicitly a second time in the previous paragraph: “We implement the diffusion path with [...] the same number of blocks as the main path, each comprising a subset of its layers (from the MLP blocks and the query layers in self-attention).”
>
> **Quality and Other Extensions**
>
> We hope our response in Claims and Evidence and the additional discussion provided some more interesting insights. Furthermore, in our latest revision, we also added new analysis and comparison of L2D with other concurrent work and traditional approaches to test-time scaling, which we hope will further strengthen our submission (please see the above response to reviewer #2 hDTm for details). Nonetheless, we hope the reviewer will let us know if there is any other specific part of our submission that they think we could further work on to improve its current quality.

---

> > ### Comment · Reviewer_hLPT · 2025-04-08
> >
> > I thank the authors for their explanations. My questions are generally well answered. I will keep my score to recommend acceptance.

---

### Official Review · Reviewer_4k8t · 2025-03-13

**Overall Recommendation:** 3

**Summary:**

This paper provided a novel perspective that treats language mode(LM) as a one-step diffusion model(DM). Thus, it proposes increasing the number of diffusion steps to boost the average score of the language model in test-time compute scaling. The methods show significant improvement in LMs in math, coding, and other reasoning.

## update after rebuttal
I have read through the author's rebuttal and confirmed that the methods are novel, However, I remain concerned that they have some performance issues that do not outperform the LoRA fine-tune baseline, indicating potential further improvement. Thus I keep my current score as Weak Accept.

**Claims And Evidence:**

The paper presents a well-articulated claim supported by strong evidence across various tasks, demonstrating clear reasoning. It highlights significant performance improvements in both tasks for recently open-sourced language models (LMs), spanning both small-scale models (1B, 1.5B) and medium-scale models (7B, 8B).

**Essential References Not Discussed:**

There are no essential references missing.

**Experimental Designs Or Analyses:**

The experimental design and analysis are sound and valid, leveraging well-established benchmarks and solid base models. The experiments are well-structured, with clear improvements demonstrated over strong baselines. Additionally, the ablation studies are well-executed, providing insights into the contributions of different components. There are no major concerns regarding the validity of the experimental setup.

**Methods And Evaluation Criteria:**

The proposed methods and evaluation criteria are well-aligned with the problem and application. The methodology is clearly defined and builds upon well-established approaches, such as language models (LMs) and diffusion models (DMs). The key novelty lies in treating LMs as a one-step DM and integrating them within a fine-tuning framework. The benchmarks used are widely recognized, and the improvements are evident based on well-established evaluation metrics.

**Other Comments Or Suggestions:**

## Strengths:
- Practical Impact: The paper focuses on practical effectiveness rather than theoretical novelty, making it highly relevant for real-world applications.

- Strong Empirical Results: The proposed approach shows clear and significant improvements over well-established baselines, demonstrating its effectiveness.

- Well-Designed Experiments: The experiments are robust, leveraging widely recognized benchmarks and solid base models. The ablation studies provide useful insights into the contributions of different components.

- Clear and Coherent Presentation: The methodology and findings are presented in a structured and logical manner, making it easy to follow.
## Weaknesses:
- Limited Theoretical Novelty: Since the work builds on existing theorems without introducing new theoretical advancements, its contribution is primarily practical.

**Other Strengths And Weaknesses:**

see strength in other comments

**Questions For Authors:**

It is noticeable that in Coding-MBPP and GeneralKnowledge-MMLU tasks the performance is less then the LoRA fine-tuning or initial models.  Is there any hypothesis for that?

**Relation To Broader Scientific Literature:**

The key contributions of the paper are well-situated within the broader scientific literature, building upon prior findings in both language models (LMs) and diffusion models (DMs). The work extends existing theoretical foundations by leveraging well-established theorems from previous studies but focuses on practical effectiveness rather than theoretical novelty. The novel approach of treating LMs as a one-step DM and integrating them in a fine-tuning framework aligns with recent trends in model unification and cross-paradigm learning. Additionally, the use of well-recognized benchmarks and clear improvements over prior baselines strengthen its connection to existing work, demonstrating meaningful progress in the field.

**Theoretical Claims:**

The proposed methods and evaluation criteria are well-aligned with the problem and application. The methodology is clearly defined and builds upon well-established approaches, such as language models (LMs) and diffusion models (DMs). The key novelty lies in treating LMs as a one-step DM and integrating them within a fine-tuning framework. The benchmarks used are widely recognized, and the improvements are evident based on well-established evaluation metrics.

---

> ### Author Rebuttal · Authors · 2025-03-31
>
> We would like to thank the reviewer for their feedback and the time they dedicated to our review.
>
> **Questions**
>
> “*It is noticeable that in Coding-MBPP and GeneralKnowledge-MMLU tasks the performance is less than the LoRA fine-tuning or initial models. Is there any hypothesis for that?*”
>
> While L2D achieves the best performance on 22 out of the 24 task/model combinations across all our baselines analyzed in [Table 1](https://anonymous.4open.science/r/rebuttal_l2d-4B0B/table1.png), as the reviewer correctly points out, there are two exceptions for the Qwen 7B model where its performance is second-best.
>
> In the MMLU task, the performance of L2D (71.11) fails to exceed the base Qwen 2.5 7B Instruct model (71.41). For this case, we would like to note that we designed our dataset to heavily focus on math and coding without emphasis on new real-world knowledge (preamble Section 4, Section 4.1, Appendix B), which we believe also explains why the other finetuning baselines get even lower results than the base instruct model (69.39 and 59.47).
>
> In the MBPP task, the performance of L2D (76.79) is also slightly behind the LoRA baseline (79.60). While this is not the case for any other task, or even any other model other than Qwen 7B on the same task, we believe this could be an indication that L2D should be viewed not just to replace but also to be used in combination with traditional weight finetuning. The potential of combining these approaches is also supported by our extensions results in the second section of [Table 2](https://anonymous.4open.science/r/rebuttal_l2d-4B0B/table2.png), where we show that L2D can be effectively used after full and lora finetunings of the base model with compounding benefits.
>
> Following the reviewer’s interest, we added an extended discussion to our latest revision of this work to specifically address these result instances, as detailed above.
>
> **Other extensions**
>
> In our latest revision, we also added new analysis and comparison of L2D with other concurrent work and traditional approaches to test-time scaling, which we hope will further strengthen our submission (please see the above response to reviewer #2 hDTm for details).

---

> > ### Comment · Reviewer_4k8t · 2025-04-07
> >
> > I appreciate the authors' response and find the core of the proposed methods novel and insightful. Although performance concerns persist in specific tasks, suggesting room for further refinement, they do not significantly impact my overall evaluation of the paper. Therefore, I maintain my original recommendation for acceptance and keep my current score.

---

> > > ### Author Response · Authors · 2025-04-07
> > >
> > > Thanks again for all the strong feedback and the interest in our work.
> > >
> > > Please do not hesitate to let us know in case you have any further suggestions for future revisions.

---

### Official Review · Reviewer_hDTm · 2025-03-13

**Overall Recommendation:** 4

**Summary:**

Authors provide a framework to combine the autoregressive LLM with diffusion models, to scale test-time compute in language reasoning tasks.

Diffusion models are primary designed for continuous domains, with few exceptions of categorical diffusion models, but primarily designed for continuous domain.

Authors use some clever techniques to introduce the diffusion framework in an LLM.

Experimental results on Maths and Coding reasoning tasks shows improved performance.


## update after rebuttal

My original overall recommendation was 4: accept. My original minor concern was that the authors did not compare with best-of-N sampling in the experiments. The rebuttal addressed my concern. The original assessment has not changed, since I recommended accept.

**Claims And Evidence:**

Claims of improved reasoning abilities hold on experimental benchmark.

**Essential References Not Discussed:**

NA

**Experimental Designs Or Analyses:**

The experimental design is sound and makes sense, though I have a question (added below).

**Methods And Evaluation Criteria:**

The method and the benchmark datasets make sense for the reasoning task.

**Other Comments Or Suggestions:**

In the experimental section, I see authors have not compared with other test-time inference methods, such as best-of-N sampling (it's variants). Any reason for that? Or do you think that method is not a suitable baseline?

**Other Strengths And Weaknesses:**

The paper is well-written and easy to understand. The problem is well motivated, and the experimental design is sound.

**Questions For Authors:**

Authors have not compared their method with best-of-N sampling method, any reason for that? I am curious.

Also, there is no RL post-training baseline as well.

Are these not relevant baselines? I might be wrong, so just wanted to pick authors brain on this.

**Relation To Broader Scientific Literature:**

The paper improves the "reasoning" ability of LLM using diffusion model, which are used in a lot of tools common people use everyday, so the work can have good impact.

**Theoretical Claims:**

The work is mostly empirical, and there are no theoretical claims to be verified.

---

> ### Author Rebuttal · Authors · 2025-03-31
>
> We would like to thank the reviewer for their feedback and the time they dedicated to our review.
>
> **Experiments for test-time inference**
>
> Since L2D scales inference with a separate new "diffusion path," we think it should be viewed as orthogonal to prior scaling approaches based on hand-designed heuristics and increasing generation length. To empirically confirm this, we would like to note that in the third section of [Table 2](https://anonymous.4open.science/r/rebuttal_l2d-4B0B/table2.png), we do compare with a tuned heuristic-guided token search strategy.  We believe these results not only show our method’s relative strengths but also validate how the two strategies do provide different and highly complementary benefits.
>
> Nonetheless, following the reviewer’s feedback and questions, we collected even more experiments for our latest revision of this work, which we hope will strengthen the argument that diffusion is highly complementary with prior scaling approaches acting over the space of generated tokens:
>
> **R1-style RL for reasoning**
>
> While the R1 paper [1], spurring the recent focus on reasoning from RL, was uploaded on the web on January 23rd, the same date as the abstract submission deadline for ICML 2025 - we share the reviewer’s interest to analyze how L2D properties should be viewed in comparison.
>
> Since RL training requires expensive multi-node settings (beyond the resources for the project) and appears mainly effective on very large LMs, we added results with the pre-trained DeepSeek R1 Distill Qwen 1.5B reasoning model. We used this model both as an additional baseline and as an extra base model from which to train L2D.
>
> [Please find a summary Table of the added results at this link.](https://anonymous.4open.science/r/rebuttal_l2d-4B0B/r1_results_summary.png)
>
> As the DeepSeek R1 model is trained on a recent private dataset, heavily focused on Math, we find its performance exceeds the original Qwen 1.5B Instruct model on this task category. However, we find this comes at an expected actual loss in performance on coding and general knowledge, which our L2D approach avoids. Moreover, further fine-tuning this baseline with L2D achieves the highest results on Math, even surpassing the much larger 7B and 8B non-RL models - as well as recovering a large part of the performance loss on the other tasks. In line with the other results, we believe these findings confirm that our new method should be viewed as complementary to RL reasoning.
>
> However, we note that evaluating these reasoning models distilled from RL was over 10x more expensive than vanilla L2D and did not work out-of-the-box, requiring to modify the prompts and relax the answer extraction code for compatibility with `<think>/<answer>` style responses.
>
> Finally, we extended our conclusion to mention that training L2D itself with concurrent R1-style RL methods could also be another interesting future research direction, taking inspiration from recent work in RL finetuning of diffusion models in computer vision [2, 3].
>
> **CoT scaling baselines**
>
> While some of our tasks already included CoT few-shot samples (e.g., GSM8K), following reviewer #1 (uZss) notable interest in this line of work, we made new CoT few-shot examples based on [4]. [Please find a summary of the results here](https://anonymous.4open.science/r/rebuttal_l2d-4B0B/cot_results_summary.png), and refer to reviewer #1 (uZss) response for further details.
>
> **Best of N**
>
> We believe the token search baseline could be viewed as an advanced version of best-of-N scaling, where the tuned beam-search scores are used as the metric to assess which is the best response. Instead, best-of-N using ground-truth correctness assumes access to an oracle verifier and is typically only considered for coding, where the oracle could come in the form of a compiler and a set of test cases to solve. In fact, this is precisely what the "pass@K" metric used for Humaneval/MBPP considers, for which we provided further results in [Table 12](https://anonymous.4open.science/r/rebuttal_l2d-4B0B/table12.png).
>
> Following the reviewer’s interest and feedback, we extended these results by providing pass@K scores also for the other math and general knowledge tasks using the Llama 1B model, which could be viewed as an upper bound for any critic-based inference-scaling approaches:
>
> [Please find a summary Table of the added results at this link.](https://anonymous.4open.science/r/rebuttal_l2d-4B0B/pass@k_math_gqa.png)
>
> We hope the reviewer will not hesitate to let us know if they believe it would be relevant to collect even more pass@K or other best-of-N analyses for our submission.
>
> [1] DeepSeek-R1: Incentivizing Reasoning Capability in LLMs via Reinforcement Learning, 2025.
>
> [2] Training Diffusion Models with Reinforcement Learning, 2024.
>
> [3] Diffusion Model Alignment Using Direct Preference Optimization, 2024.
>
> [4] Towards Understanding Chain-of-Thought Prompting: An Empirical Study of What Matters, 2023.

---

> > ### Comment · Reviewer_hDTm · 2025-04-07
> >
> > Thank you for your time in writing a rebuttal to my review, and thank you running additional experiments. I am convinced that best-of-N sampling is not a fair baseline to compare with.

---

> > > ### Author Response · Authors · 2025-04-07
> > >
> > > Thanks again for your time and the insightful comments.
> > >
> > > We are glad we were able to address your questions. Please do not hesitate to let us know if anything else comes to mind.

---

### Official Review · Reviewer_uZss · 2025-03-15

**Overall Recommendation:** 3

**Summary:**

The paper introduces L2D, a method that integrates the scaling properties of diffusion models into pre-trained language models (LMs) to enhance reasoning skills and computational scalability. L2D improves pre-trained LMs on math, coding, and various reasoning tasks, outperforming LoRA and full fine-tuning methods.

**Claims And Evidence:**

The main claim of the paper is that the diffusion-based fine-tuning approach enables the pre-trained language model to scale at test time for reasoning.

By interpreting LMs trained with cross-entropy loss as single-step diffusion models, this framework is naturally facilitated, clearly supporting the convincing potential for test-time scaling through multi-step processes.

**Essential References Not Discussed:**

Test-time scaling of LMs
- Large language monkeys: Scaling inference compute with repeated sampling, ArXiv, 2024
- Don't throw away your value model! Generating more preferable text with Value-Guided Monte-Carlo Tree Search decoding, COLM, 2024
- Scaling LLM Test-Time Compute Optimally Can be More Effective than Scaling Parameters for Reasoning, ICLR, 2025
- and much more on this topic

Test-time scaling of Diffusions
- Inference-Time Alignment of Diffusion Models with Direct Noise Optimization, ArXiv, 2024
- Inference-Time Alignment in Diffusion Models with Reward-Guided Generation: Tutorial and Review, ArXiv, 2025
- Test-time Alignment of Diffusion Models without Reward Over-optimization, ICLR, 2025

**Experimental Designs Or Analyses:**

The experiments are generally well-designed, including tasks, data, and metrics, but the baselines sufficient for verifying the efficacy of the proposed method are not adequately compared.

**Methods And Evaluation Criteria:**

The method is technically sound.

The reviewer's main concerns lie in the evaluation.
- Lack of CoT-like LM baselines. In experiments, there are no CoT-like test-time scaling baselines for LMs. L2D should be compared with these baselines to prove its genuine effectiveness on test-time scaling.
- Lack of post-training baselines for reasoning. LoRA and full fine-tuning LMs on reasoning datasets seem not enough as fine-tuning baselines for reasoning tasks. More decent baselines using RL post-training for reasoning are essential to validate the effectiveness of L2D.

**Other Comments Or Suggestions:**

Typo: Eq (2) might be L^{L2D} ~

**Other Strengths And Weaknesses:**

The reviewer's main concern is the lack of in-depth experimental and theoretical analysis regarding the advantages and disadvantages of the proposed method compared to existing LM reasoning frameworks.

**Questions For Authors:**

Can diffusion fine-tuning be done using LoRA?

**Relation To Broader Scientific Literature:**

This work may have an impact by connecting the literature of LMs and diffusion through fine-tuning LMs with the diffusion framework. However, in my opinion, it has not been thoroughly examined how integrating diffusion into LM fine-tuning has an advantage over the existing works on test-time scaling in traditional LMs.

**Theoretical Claims:**

The proposed method is based on the diffusion formulation, but there are no theoretical claims in the paper that guarantee or analyze the efficacy or stability of the proposed approach.

---

> ### Author Rebuttal · Authors · 2025-03-31
>
> We would like to thank the reviewer for their feedback and the time they dedicated to our review.
>
> **Experiments**
>
> We added clarifications and experiments to our revised work to address the reviewer’s concerns. We hope these will strengthen the argument that L2D is a novel orthogonal method, highly complementary with scaling approaches based on increasing generation length.
>
> **CoT scaling baselines**
>
> Following the reviewer’s feedback on CoT prompting, we made versions of our tasks with new CoT few-shot examples designed to elicit better and longer reasoning. In particular, these examples were obtained by prompting Claude Sonnet 3.7 to provide effective CoT based on the heuristics proposed in [1]. We note this change significantly increased inference time, especially for our multiple-choice tasks, going from the models generating a single letter answer directly to producing lengthy reasonings beforehand (averaging 84 new tokens).
>
> [Please find a summary Table of the added results at this link.](https://anonymous.4open.science/r/rebuttal_l2d-4B0B/cot_results_summary.png)
>
> As shown, this tuned CoT prompting strategy indeed achieves improvements for both the base Llama model and our other finetuning baselines, albeit lower than our previous baseline results scaling test-time compute with token search (third section of [Table 2](https://anonymous.4open.science/r/rebuttal_l2d-4B0B/table2.png)) and L2D. Furthermore, in line with our other findings, using L2D models together with CoT prompting yields compounding test-time benefits, which we believe evidences the synergy between our method and this orthogonal approach.
>
> **R1-style RL for reasoning**
>
> While the R1 paper [2], spurring the recent focus on reasoning from RL, was uploaded on the web on January 23rd, the same date as the abstract submission deadline for ICML 2025 - we understand the importance of considering this relevant line of work.
>
> Since RL training requires expensive multi-node settings (far beyond L2D and the resources for the project) and appears mainly effective on very large LMs, we added results with the pre-trained DeepSeek R1 Distill Qwen 1.5B reasoning model. We not only used this model as an additional baseline but also as an extra base model from which to train L2D.
>
> [Please find a summary Table of the added results at this link.](https://anonymous.4open.science/r/rebuttal_l2d-4B0B/r1_results_summary.png)
>
> As the DeepSeek R1 model is trained on a private dataset, heavily focused on Math, we find its performance exceeds the original Qwen 1.5B Instruct model on this task category. However, we find this comes at an expected actual loss in performance on coding and general knowledge, which our L2D approach avoids. Moreover, further fine-tuning this baseline with L2D achieves the highest results on Math, even surpassing the much larger 7B and 8B non-RL models, and recovers a large part of the performance loss on the other tasks. In line with the other results combining L2D with other traditional test-time scaling approaches, we believe these findings suggest that our new method should be viewed as complementary also to RL reasoning.
>
> However, we note that evaluating these reasoning models distilled from RL was over 10x more expensive than vanilla L2D and did not work out-of-the-box, requiring to modify the prompts and relax the answer extraction code for compatibility with `<think>/<answer>` style responses.
>
> Finally, we extended our conclusion to mention that training L2D itself with concurrent R1-style RL methods could also be another interesting future research direction, taking inspiration from recent work in RL finetuning of diffusion models in computer vision [4, 5].
>
> **Related work**
>
> We would like to thank the reviewer for providing us with additional references to the related literature on test-time scaling of LMs and diffusion. We have included all their suggestions and also connections to the concurrent RL-based line of research (e.g., [2, 3]) in the new revision of our work.
>
> **Questions**
>
> As described in Sections 3.1 and 4.1, all our main implementations used for the [Table 1 results](https://anonymous.4open.science/r/rebuttal_l2d-4B0B/table1.png) are already optimizing the diffusion path of L2D with LoRA, which is precisely what allows L2D to be over an order of magnitude more parameter-efficient than full weight finetuning. While optimizing all L2D parameters appears to further increase performance, especially on coding ([Table 2](https://anonymous.4open.science/r/rebuttal_l2d-4B0B/table2.png)), it comes with non-negligible additional costs comparable to the ones between traditional LoRA and full weight finetunings.
>
> [1] Towards Understanding Chain-of-Thought Prompting: An Empirical Study of What Matters, 2023.
>
> [2] DeepSeek-R1: Incentivizing Reasoning Capability in LLMs via RL, 2025.
>
> [3] s1: Simple test-time scaling, 2025.
>
> [4] Training Diffusion Models with RL, 2024.
>
> [5] Diffusion Model Alignment Using DPO, 2024.

---

> > ### Comment · Reviewer_uZss · 2025-04-08
> >
> > Thank you for your rebuttal. I appreciate the inclusion of comparisons with the CoT baseline and the experimental results for the R1-style reasoning model. These have comprehensively addressed all my questions. I will update my rating accordingly. I hope that our discussion is well reflected in the final version.

---

### Decision · Program_Chairs · 2025-05-01

**Decision:**

Accept (poster)

**Comment:**

The reviews and discussions on the manuscript highlighted both strengths and weaknesses.

On the one hand, the manuscript clearly contributes to the study of large language models, from an approach via diffusions to a finetuning framework by integrating guidance techniques to achieve increased accuracy in answer questions on specific topics. The paper has been identified as mainly empirical, showing experiments with various models and benchmarks.

On the other hand, the paper required some revisions. As commented by reviewers, some comparisons were missing (e.g. with best-of-N sampling and comparison between the CoT reasoning baseline and the latest R1 reasoning model). There was a comment raised that it looks like in some task this work does not out-perform simple LoRA baseline. However, the rebuttals alleviated some concerns and the reviewers reiterated their mainly positive evaluation of this paper.